**A new parameterization scheme of the real part of the ambient urban aerosols refractive index**
Gang Zhao[1], Tianyi Tan[2], Weilun Zhao[1], Song Guo[2], Ping Tian[3], Chunsheng Zhao[1*]
1 Department of Atmospheric and Oceanic Sciences, School of Physics, Peking University, Beijing,
China
2 State Key Joint Laboratory of Environmental Simulation and Pollution Control, College of
Environmental Sciences and Engineering, Peking University, Beijing 100871, China
3 Beijing Key Laboratory of Cloud, Precipitation and Atmospheric Water Resources, Beijing 100089,
China
*Correspondence to: Chunsheng Zhao (zcs@pku.edu.cn)
**Abstract**
The refractive index of ambient aerosols, which directly determines the aerosol optical properties,
is widely used in atmospheric models and remote sensing. Traditionally, the real part of the refractive
index (RRI) is parameterized by the measurement of ambient aerosol main inorganic components. In
this paper, the characteristics of the ambient aerosol RRI are studied based on the field measurement
in the East China. The results show that the measured ambient aerosol RRI varies significantly between
1.36 and 1.56. The direct aerosol radiative forcing is estimated to vary by 40% when the RRI were
varied between 1.36 and 1.56. We find that the ambient aerosol RRI is highly correlated with the
aerosol effective density ($\rho_{eff}$) rather than the main chemical components. However, parameterization
of the ambient aerosol RRI by $\rho_{eff}$ are not available due to the lack of corresponding simultaneous
field measurements. For the first time, the size-resolved ambient aerosol RRI and $\rho_{eff}$ are measured
simultaneously by our designed measurement system. A new parameterization scheme of the ambient
aerosols RRI using $\rho_{eff}$ is proposed for the urban environments. The measured and parameterized
RRI agree well with the correlation coefficient of 0.75 and slope of 0.99. Knowledge of the ambient
aerosol RRI would improve our understanding of the ambient aerosol radiative effects.
**1 Introduction**
Atmospheric aerosols can significantly influence the regional air quality (An et al., 2019;Zhang et
al., 2015). They change the    climate system by scattering and absorbing the solar radiation (Seinfeld
et al., 1998;Wang et al., 2013). However, estimation of the aerosol radiative effects remains large
uncertainties due to the high temporal and spatial variations in aerosol microphysical properties
(Levoni et al., 1997). The complex refractive index (RI), which directly determines the aerosol
scattering and absorbing abilities (Bohren and Huffman, 2007), is one of the most important
microphysical parameters of aerosol optics and radiation. RI is widely employed in atmospheric
models and remote sensing (Zhao et al., 2017). When estimating the direct aerosol radiative forcing
(DARF), many studies showed that great uncertainties may arise due to small uncertainties in the real
part of the RI (RRI). It was found that a small perturbation in RRI (0.003) can lead to an uncertainty
of 1% in DARF for non-absorbing particles (Zarzana et al., 2014). An increment of 12% in the DARF
occurred when the RRI increased from 1.4 to 1.5 (Moise et al., 2015) over the wavelength range
between 0.2 μm and 5 μm. Therefore, it is necessary to measure or parameterize the ambient aerosol
RRI with high accuracy.
Traditionally, the RRI is derived from measurements of aerosol main inorganic chemical
compositions    (Han et al., 2009). For the ambient aerosol with multiple components, linear volume
average of known aerosol chemical composition is widely used to estimate the aerosol effective *RRI*
(Hand and Kreidenweis, 2002;Liu and Daum, 2008;Hänel, 1968;Wex et al., 2002) with :

$$RRI_{eff} = \sum_i (f_i \cdot RRI_i) \qquad (1)$$

Where $f_i$ and $RRI_i$ are the volume fraction and real part of refractive index of known composition
*i*. However, the influences of organic component on the aerosol RRI were not considered when
estimating the RRI using the traditional method. The organic component contributes more than 20%
of the total aerosol component in China (Hu et al., 2012;Liu et al., 2014). At the same time, RRI of the
organic aerosol changes significantly between 1.36 and 1.66 (Moise et al., 2015). Ignoring the organic
component may lead to significant biases when estimating the ambient aerosol RRI. The comparison
between the estimated RRI using main aerosol composition and measured aerosol RRI using other
method was not available due to the lack of measurement of ambient aerosol RRI.

Information of RRI may be helpful for the knowledge of ambient aerosol chemical information.

Many studies find that ambient aerosols of different size have different properties such as shape (Peng
et al., 2016), chemical composition (Hu et al., 2012) and density (Qiao et al., 2018). Up until now,
there is limit information about the size-resolved RRI ($\widetilde{RRI}$, denoted in Table. S1) of ambient particles.
Characteristics of the ambient aerosol $\widetilde{RRI}$ were not well studied yet.

The RRI of mono-component particle is defined as (Liu and Daum, 2008):

$$\frac{RRI^2-1}{RRI^2+2} = \frac{N_A\alpha}{3M}\rho_{eff} \qquad (2)$$
where $N_A$ is the universal Avagadro's number, $\alpha$ is the mean molecular polarizability, M is the
molecular weight of the material and $\rho_{eff}$ is the mass effective density of the chemical component.
The RRI should be highly related to $\rho_{eff}$. However, there was no study that investigated the
relationship between the RRI and $\rho_{eff}$ of ambient aerosol in China.

The $\rho_{eff}$ of ambient aerosols is one of the crucial parameters in aerosol thermo-dynamical and

optical models. It can be used to infer the ambient particle aging process (Peng et al., 2016). Based on
equation 2, the aerosol $\rho_{eff}$ is directly related to the aerosol RRI. Few studies measure the ambient
aerosol RRI and $\rho_{eff}$ simultaneously. So far, parameterizations of the RRI by $\rho_{eff}$ using the
simultaneous measurements are not available. Real-time measurements of the $\rho_{eff}$ and aerosol RRI
concurrently can help to better understand the relationship between the aerosol RRI and $\rho_{eff}$.

In this study, the aerosol $\widetilde{RRI}$ and size resolved $\rho_{eff}$ ($\widetilde{\rho_{eff}}$) are measured simultaneously during

a field measurement conducted in Taizhou in the East China. The ambient aerosol $\widetilde{RRI}$ is measured
by our designed system, which combines a differential mobility analyzer (DMA) and a single particle
soot photometer (SP2) (Zhao et al., 2019). The $\widetilde{\rho_{eff}}$ is measured by using a centrifugal particle mass
analyzer (CMPA) and a scanning mobility particle sizer (SMPS). The characteristic of the $\widetilde{RRI}$ and
$\widetilde{\rho_{eff}}$ are analyzed in this study. It is the first time that the $\widetilde{RRI}$ and $\widetilde{\rho_{eff}}$ of the ambient aerosol are
measured simultaneously. A parameterization scheme of the RRI by the $\rho_{eff}$ using the simultaneous
measurement is proposed. Based on the measured variability of the measured RRI, we estimated the
corresponding variation of the aerosol direct aerosol radiative forcing, which to some extent give
valuable knowledge for the influence of aerosol RRI variations on aerosol radiative effects.

The structure of this study is as follows: the descriptions of the instrument setup is given in section

2.1, 2.2 and 2.3. The methodology of evaluating the aerosol optical properties and radiative effects
corresponding to the variations of the measured RRI are shown in section 2.4 and 2.5 respectively.
Section 3.1 describes the characteristics of the measured $\widetilde{RRI}$ and $\widetilde{\rho_{eff}}$. Section 3.3 proposes the
parameterization of the aerosol RRI. The corresponding variations in aerosol optical properties and
radiative effects corresponding to the variations of the measured RRI are both discussed in section 3.4.
**2 Data and Methods**
**2.1 Description of the measurement campaign**

The measurement was conducted in a suburban site Taizhou (119$^\circ$57'E, 32$^\circ$35'N), as shown in

fig. 1(a), which lies in the south end of the Jianghuai Plain in the central Eastern China. It is located
on the north east of the megacity Nanjing with a distance of 118 km. Another megacity Shanghai is
200 km away from Taizhou in the southeastern direction. The industrial area between Nanjing and
Shanghai has experienced severe pollutions in the past twenty years. The average Moderate Resolution
Imaging Spectroradiometer (MODIS) aerosol optical depth data at 550nm over the year 2017, as
shown in fig. 1(b), also reflects that the measurement site is more polluted than the surrounding areas.
During the field campaign, all of the instruments were placed in a container, in which the temperature
was well controlled within 24±2 $^\circ$C. The sample air was collected from a PM$_{10}$ impactor (Mesa Labs,
Model SSI2.5) mounted on the top of the container and then passed through a Nafion dryer tube to
ensure that the relative humidity of the sample particles was controlled below 30%.
Along with the measurement of the $\widetilde{RRI}$ and $\widetilde{\rho_{eff}}$, the aerosol scattering coefficients ($\sigma_{sca}$) at three
different wavelengths (450, 525 and 635 nm) were measured by an nephelometer (Aurora 3000,
Ecotech, Australia) (Müller et al., 2011) at a resolution of 5 minutes. The scattering truncation and
non-Lambertian error was corrected using the same method as that of Ma et al. (2011). The aerosol
water-soluble ions ($NH_4^+$, $SO_4^{2-}$, $NO_3^-$, $Cl^-$) of $PM_{2.5}$ were measured by an In situ Gas and Aerosol
Compositions Monitor (TH-GAC3000, China). The mass concentration of elementary carbon and
organic carbon (OC) of PM2.5 were measured using a thermal optical transmittance aerosol carbon
analyzer (ECOC, Focused Photonics Inc.). The concentrations of Organic matters (OM) are achieved
through multiplying OC concentration by 1.4 (Hu et al., 2012). The time resolution of the aerosol
composition measurement was one hour.
Another field measurement were conducted in the campus of Peking University (PKU) (N39°59′,
E116°18′), in North China Plain, where the aerosol effective density and real part of the refractive
index are measured concurrently from 16[th] to 20[st], December in 2018. More detail of this site can refer
to (Zhao et al., 2018).

## 113 2.2 Measuring the $\widetilde{RRI}$

A coupling DMA-SP2 system was employed to measure the aerosol $\widetilde{RRI}$ from 24[th], May to 18[th],
June in 2018. This system is introduced elsewhere by (Zhao et al., 2019) and a brief description is
presented here. As schematically shown in fig. 2, the monodispersed aerosols selected by a DMA
(Model 3081, TSI, USA) are drawn into a SP2 to measure the corresponding scattering properties. The
SP2 is capable of distinguishing the pure scattering aerosols from the black carbon (BC) containing
aerosols by measuring the incandescence signals at 1064 nm. For the pure scattering aerosol, the
scattering strength (S) measured by SP2 is expressed as:
$$S = C \cdot I_0 \cdot (\sigma_{45°} + \sigma_{135°}) \qquad (3),$$
where $C$ is a constant that is determined by the instrument response character; $I_0$ is the instrument's
laser intensity; $\sigma_{45°}$ and $\sigma_{135°}$ is the scattering function of the sampled aerosol at 45° and 135°,
respectively;. From Mie scattering theory, aerosol size and RRI directly determine the scattering
function at a given direction. Inversely, the aerosol RRI can be retrieved when the aerosol size and
scattering strength are determined. This system can measure the ambient aerosol $\widetilde{RRI}$ with uncertainty
less than 0.02 (Zhao et al., 2019).
Before the measurement, this system is calibrated with ammonia sulfate (RRI=1.52). After
calibration, ammonium chloride is used to validate the method of deriving the RRI at different aerosol
diameters. The RRI value of ammonium chloride is 1.642 (Lide, 2006) and the measured RRI of
ammonium chloride is in the range between 1.624 and 1.656 in our study. Therefore, this measurement
system can measure the ambient aerosol RRI with high accuracy.
**2.3 Measuring the $\widetilde{\rho_{eff}}$**
The $\widetilde{\rho_{eff}}$ is measured by a Centrifugal Particle Mass Analyzer (CPMA, version 1.53, Cambustion
Ltd, UK) in tandem with a Scanning Mobility Particle Sizer (SMPS) system from 12[th], June to 18[th],
June in 2018. The $\rho_{eff}$ is defined as
$$\rho_{eff} = \frac{m_p}{\frac{\pi}{6} \times d_m^3} \qquad (4),$$
Where $m_p$ is the particle mass and $d_m$ is the aerosol mobility diameter selected by DMA.
The controlling of the CPMA-SMPS system is achieved by self-established Labview software.
The CPMA is set to scan twelve different aerosol mass at 1.0, 1.4, 2.0, 2.9, 4.2, 5.9, 8.5, 12.1, 17.2,
24.6, 35.0 and 50.0 fg every five minutes respectively. The SMPS scan the aerosol diameters between
60nm and 500nm every 5 minute, which results in a period of one hour for measuring the effective
density of different mass.
At the beginning of the field measurement, the CPMA-SMPS system is calibrated using the PSL
particles with different mass. The corresponding measured effective densities of PSL particles are 1.04
and 1.07 g/cm$^3$, which agree well with the PSL material density of 1.05 g/cm$^3$.
Fig. 3 gave three examples of the aerosol PNSDs that passed the CPMA and were measured by
the SMPS. The mass values of the aerosol that can pass through the CPMA were set to be 12, 1 and
1.4 fg respectively. From fig. 3, these aerosols that pass through the CPMA were mainly composed of
three modes. For each mode, the aerosol number concentrations were fit by log-normal distribution
function:
$$N(H) = \frac{N_0}{\sqrt{2\pi}\log(\sigma_g)} \cdot exp\left[-\frac{\log Dp - \log(Dp)}{2\log^2(\sigma_g)}\right], \qquad (5)$$
where $\sigma_g$ is the geometric standard deviation; $Dp$ is the geometric mean diameter and $N_0$ is the
number concentrations for a peak mode. The geometric mean diameter is further analyzed.
We would demonstrate the mode 1, 2 and 3 in fig. 3 correspond to those aerosols of absorbing
aerosol, scattering aerosol, and scattering aerosol with double charges respectively.
Based on the principle of CPMA, when the CPMA is selecting the aerosols at mass $m_0$ of single
charged aerosol particles. Theses multiple-charged (numbers of charges is $n$) aerosol particles with
mass concentration of $nm_0$ can pass through the CPMA at the same time. We assumed that the
geometric diameter of the single charge aerosol particles was $D_0$, and the effective density among
different aerosol diameter didn't have significant variations. Thus, the geometric diameter of the
multiple charged aerosol particles is $\sqrt[3]{n}$m.
As for the DMA, when a voltage ($V$) is applied to the DMA, only a narrow size range of aerosol
particles, with the same electrical mobility ($Z_p$) can pass through the DMA (Knutson and Whitby,
1975). The $Z_p$ is expressed as:
$$Z_P = \frac{Q_{sh}}{2\pi VL} ln(\frac{r_1}{r_2}) \qquad (6)$$

where $Q_{sh}$ is the sheath flow rate; $L$ is the length of the DMA; $r_1$ is the outer radius of annular space
and $r_2$ is the inner radius of the annular space. The aerosol $Z_p$, which is highly related to the aerosols
diameter ($D_p$) and the number of elementary charges on the particle ($n$), is defined as:
$$Z_p = \frac{neC(D_p)}{3\pi\mu D_p} \qquad (7)$$

where $e$ is the elementary charge; $\mu$ is the gas viscosity coefficient, $C(D_p)$ is the Cunningham slip
correction that is defined by:
$$C = 1 + \frac{2\tau}{D_p}(1.142 + 0.558e^{-\frac{0.999D_p}{2\tau}}) \qquad (8)$$

where $\tau$ is the gas mean free path.
Therefore, the electrical diameter Zp(n) of the particles with n charges and diameters $\sqrt[3]{n}$m can
be calculated based on equation 5. Thus, the corresponding diameter (Dn) measured by the DMA can
be calculated with electrical diameter Zp(n) and single charged particle by using equation 5 again. The
relationship of the Dn and the aerosol diameter selected by the DMA can be determined by changing
the aerosol Dp and charge numbers. The results were shown in fig. 4.
The fit geometric diameters of mode 2 and mode 3 were also shown in fig. 4. From fig. 4, the
measured diameter relationships of the mode 2 and mode 3 agree well with the calculated one between
the single charged and double charged diameters. The little deviation might result from the
assumptions that the aerosol effective density doesn't change among different diameters. We
concluded that the mode 3 corresponds to the double-charged aerosols. Mode 3 is not used in our study.
Mode 1 and mode 2 corresponding to the effective densities around 1.0 $g/cm^3$ and 1.5 $g/cm^3$.
Previous studies have shown that the ambient BC aerosol was chain like in the morphology and had
smaller effective density values (Peng et al., 2016). At the same time, the fit aerosol number
concentrations of mode one is only between 1/5 to 1/3 of the mode two. Based on the size-selected
aerosol properties measured by the SP2, there were only mean 25% percent of the ambient aerosols
that contain BC. Therefore, the mode 1 and mode 2 corresponded to the BC-contained aerosols and
scattering aerosols respectively. There were some compacted BC-contained aerosols that may fit in
mode 2. We focus on the fit geometric mean diameter of mode 2, which corresponding to these
scattering aerosols that dominated this mode. Therefore, these compacted BC aerosols would not
influence our final conclusion.
The effective density used in our study correspond to the geometric diameters of mode 2. Thus,
both the measured aerosol $\rho_{eff}$ and RRI correspond to these scattering aerosols
**2.4 Calculate aerosol optical properties using different RRI**
The aerosol optical properties are highly related to the RRI. From Mie scattering theory, the variation
in aerosol RRI may result in significant variations in the aerosol optical properties, such as aerosol
extinction coefficient ($\sigma_{ext}$), the $\sigma_{sca}$, the single scattering albedo (SSA), and the asymmetry factor
(g) (Bohren and Huffman, 2007). The $\sigma_{ext}$, SSA and g are the most important three factors that
influence the aerosol radiative properties in radiative calculation (Kuang et al., 2015;Zhao et al., 2018).
In this study, the sensitivity studies of the aeorsol optical proprties to the aerosol RRI are carried
out by employing the Mie scattering theory. The input variables of Mie scattering model includes the
aerosol particle number size distriubiton (PNSD) and BC mixing state and aerosol complex refractive
index. The Mie model can calculate the $\sigma_{ext}$, $\sigma_{sca}$, SSA and g. The mixing state of the ambient BC
comes from the measurements of the DMA-SP2 system. All of the aerosols are divided into pure
scattering aerosols and BC-containing aerosols. The BC-containing aerosols are assumed to be core-
shell mixed. As for the RI of BC, 1.8+0.54i is used (Kuang et al., 2015). With this, the aerosol $\sigma_{ext}$,
$\sigma_{sca}$, SSA and g at different RRI values can be calculated.
**2.5 Estimating the aerosol DARF**
In this study, the DARF under different aerosol RRI conditions is estimated by the Santa Barbara
DISORT (discrete ordinates radiative transfer) Atmospheric Radiative Transfer (SBDART) model
(Ricchiazzi et al., 1998). Under the cloud-free conditions, DARF at the TOA is calculated as the
difference between radiative flux under aerosol-free conditions and aerosol present conditions
(Kuang et al., 2016). The instant DARF value is calculated over the wavelength range between 0.25
μm and 4 μm.
Input data for the model are shown below. The vertical profiles of temperature, pressure and water
vapor, which adopt the radiosonde observations at Taizhou site. The measured mean results
corresponding the field measurement period are used. Vertical distributions of aerosol $\sigma_{ext}$, SSA and
g with a resolution of 50 m, are resulted from the calculation using the Mie Model and parameterized
aerosol vertical distributions. Methods for parameterization and calculation of the aerosol optical
profiles can be found in Zhao et al. (2018). The surface albedo adopt the mean results of MODIS V005
Climate Modeling Grid (CMG) Albedo Product (MCD43C3) at the area of Taizhou from May, 2017
to April, 2018. The other default values are used in the simulation (Ricchiazzi et al., 1998).
**3 Results and Discussions**
**3.1 The Measurements Results**
The overview of the measurement is shown in fig. 5. During the measurement, the $\sigma_{sca}$ is relatively
low with a mean value of 167±74 Mm[-1]. There were one major pollution episodes occurred based on
the $\sigma_{sca}$ time series as shown in fig. 5(a). This pollution happens on 13[th], June and doesn't last long.
The corresponding $\sigma_{sca}$ reaches 540 Mm[-1]. A moderate polluted condition between 14[th], June and
15[th], June is observed. The aerosol PNSD changes substantially with the pollution conditions as
shown in fig. 5(b). The geometric median aerosol diameter changes between 30 nm and 105 nm. The
median diameter tends to be lower when the surrounding is cleaner. Despite the median diameter
reaches 105 nm on 16[th], June, the surrounding is relative clean due to the low aerosol number
concentration. The $\widetilde{RRI}$ varies from 1.34 to 1.54 and the $\widetilde{\rho_{eff}}$ ranges between 1.21 to 1.80 g/cm$^3$ as
shown in fig. 5 (c) and (d). From fig. 5, the measured RRI shows the same variation pattern with the
$\rho_{eff.}$ Both the $\widetilde{RRI}$ and $\widetilde{\rho_{eff}}$ increase with the diameter, which may indicate that the aerosol chemical
composition varies among different aerosol particle size.

As for the $\widetilde{RRI}$, the corresponding mean RRI values for aerosol diameter at 200 nm, 300 nm and

450 nm are 1.425±0.031, 1.435±0.041, 1.47±0.059. When comparing the probability distribution of
the RRI for different diameter in fig. 6, the RRI is more dispersed when the particle size increases,
implicating that the aerosol compositions become complicated when the aerosol get aged. Fig. 6 (a),
(c) and (e) give diurnal variation of the $\widetilde{RRI}$ values at different particle sizes of 200 nm, 300 nm and
450 nm. The RRI shows diurnal cycles for different diameters. They reach the peak at about 15:00 in
the   afternoon and fall to the valley at around 9:00 in the morning.
The range of the measured RRI (1.34~1.56) is a little wider than the literature values. The past
measurement of the ambient aerosol RRI values varies between 1.4 and 1.6 (Dubovik, 2002;Guyon et
al., 2003;Zhang et al., 2016) over different measurement site. This is the first time that such high
variations in ambient aerosol RRI were observed at one site.
The $\widetilde{\rho_{eff}}$ shows almost the same diurnal variations as the $\widetilde{RRI}$ as shown fig. S1. The diurnal
variations of the $\widetilde{\rho_{eff}}$ is more dispersed because the time period of measuring the $\widetilde{\rho_{eff}}$ is shorter (7
days) comparing with the time of $\widetilde{RRI}$ (28 days). It is evident that the $\rho_{eff.}$ increased with particle
size. The difference of $\rho_{eff}$ among different particle size should be resulted from different
contributions of chemical compositions, especially the OM. Based on the previous measurement of the
size-resolved chemical compositions using a micro orifice uniform deposit impactors (MOUDI), the
mass fraction of OM get decreased with the increment of aerosol diameter (Hu et al., 2012). At the
same time, the effective density of OM is lower than the other main inorganic compositions.
**3.2 Aerosol Chemical Composition versus the RRI**
From equation (1) and (2), the aerosol RRI can be determined by aerosol chemical composition (Liu
and Daum, 2008). Many studies calculate the RRI using the measurement results of the relative
contributions of aerosol chemical composition (Yue et al., 1994;Hänel, 1968;Guyon et al.,
2003;Stelson, 1990;Wex et al., 2002). However, there is no comparison between the RRI calculated
from chemical composition and real-time measurement one until now. In this study, the relationship
between the measured RRI and the mass fraction of each ion components is investigated.
As illustrated in fig. 7, The measured RRI have implicit relationship with the mass fraction of the
$\sigma_{sca}$ at 525 nm, OM, $SO_4^{2-}$, $Cl^-$, and $NO_3^-$. The mass ratio of $NH_4^+$ seems to be negatively
correlated with the measured RRI. At the same time, the measured RRI values have no clear
relationship with the absolute mass concentrations of the main aerosol chemical components, as shown
in fig. S2.
The RRI is also calculated by applying the method proposed by Stelson (1990), in which the bulk
chemical composition is used. The comparison between the calculated RRI and the measured RRI is
shown in fig. 8. It can be noticed that the calculated RRI and the measured RRI doesn't agree well.
There are several reasons that may cause the discrepancies. The first reason might be that the aerosol
chemical information used in the method is the average mass of whole aerosol population. The aerosol
chemical composition may vary significantly among different size. Secondly, the OM of the ambient
aerosols is very complicated and the influence of the OM on the aerosol RRI has not been studied well.
Therefore, more research is necessary when parameterizing the ambient aerosol RRI with the measured
aerosol chemical composition.
We would demonstrate that the measured RRI at a given diameter of 300 nm is in consistent with
that of the bulk aerosol optical properties derived $RRI_{opt}$. The aerosol-effective RRI of bulk aerosol
was retrieved by applying the Mie scattering theory to the aerosol particle number size distribution
(PNSD), aerosol bulk scattering coefficient and aerosol absorbing coefficient data (Cai et al., 2011).
Fig. 9 shows the time series of measured RRI and retrieved $RRI_{opt}$. Results in fig. 9 show that the
measured $RRI$ and retrieved $RRI_{opt}$ shows good consistence with $R^2 = 0.58$. Therefore, the measured
size-resolved aerosol RRI can be used to represent the bulk aerosol optical properties. The measured
RRI at 300 nm and calculated aerosol RRI using the bulk aerosol main chemical composition should
to some extent correlated with each other. However, as shown in fig. 8, the measured RRI at 300 nm
and calculated RRI using the method of Stelson (1990) has $R^2$ of 0.07. Therefore, calculating the
ambient aerosol RRI calculated using bulk aerosol main inorganic component may lead to great
uncertainties.

**3.3 Parameterizing the RRI using $\rho_{eff}$**

As shown in fig. 5, there is good consistence between the variation of the measured $\widetilde{RRI}$ and $\widetilde{\rho_{eff}}$.
When defining the specific refractive index Re with $Re = \frac{RRI^2-1}{RRI^2+2}$, we found that the Re is highly
correlated with $\rho_{eff}$ by a $R^2$ equaling 0.75 and slope 0.99 (fig. 10). The linear relationships between
the Re and $\rho_{eff}$ is:

$$\frac{RRI^2-1}{RRI^2+2} = 0.18\rho_{eff} \qquad (9).$$

The RRI can be calculated based on equation 6:

$$RRI = \sqrt{\frac{1+0.36\rho_{eff}}{1-0.18\rho_{eff}}} \qquad (10).$$

Based on equation 9 and fig. 10 the aerosol RRI can be parameterized by the $\rho_{eff}$ with high accuracy
and the uncertainties of the calculated RRI using equation 10 can be constrained within 0.025. The
aerosol $\rho_{eff}$ is easier to be measured, and equation 10 might be used as a good probe of parameterizing
the RRI.
The RRI were also calculated using the parameterization scheme equation 9 with the field
measurement data at PKU site. The slope and correlation coefficient at PKU site are 0.97 and 0.56
respectively. The calculated and measured RRI show good consistence. Therefore, this scheme is
applicable for different seasons at both Center China and North China Plain. We also compared the
measured RRI and calculated RRI using the measured $\rho_{eff}$ that have been previously published
(Hänel, 1968;Tang and Munkelwitz, 1994;Tang, 1996;Hand and Kreidenweis, 2002;Guyon et al.,
2003). The measured and calculated RRI show good consistence with $R^2$ of 0.91 and slope of 1.0.
Therefore, our parameterization scheme is universal and applicable for the urban aerosols.
This parameterization scheme is easy to use because the effective density is the only parameter used
as input. We have demonstrated that the traditional method of calculating the RRI using aerosol main
chemical components can have significant bias because the effects of organic aerosol is not considered.
The RRI can be easy to calculate based on our parameterization scheme, as the effective density of
ambient aerosol is rather easier to measure.
In the previous, Liu and Daum (2008) summarized some of the measured RRI and the $\rho_{eff}$, and
parameterized the RRI as
$$\frac{RRI^2 - 1}{RRI^2 + 2} = 0.23\rho^{0.39} \qquad (11).$$
The feasibility of this scheme is tested here and the results are shown in fig. 8. The measured and
parameterized RRI using the method of Liu and Daum (2008) deviated from 1:1 line. The relationship
of the effective density and RRI were mainly from 4000 pure materials and few ambient aerosol data.
However, the ambient aerosol were far from pure materials. At the same time, most of the pure
materials have negligible contribution to the total aerosol. Therefore, the parameterization scheme
from Liu and Daum (2008) can't well describe the relationships of the effective density and RRI of
ambient aerosol.
**3.4 Influence of RRI Variation on Aerosol Optical Properties and Radiative Properties**
The measured RRI varies between 1.34 and 1.56 during the field campaign. The corresponding
aerosol optical properties are estimated. When estimating the aerosol optical properties with different
aerosol RRI, the measured mean aerosol PNSD and mixing states are used. Fig. 11 gives the variation
of the aerosol $\sigma_{sca}$, SSA and g. From fig. 11, the $\sigma_{sca}$ varies from 162 Mm$^{-1}$ to 308 Mm$^{-1}$. The SSA
varies between 0.843 and 0.895, which matches the variations of the dry aerosol SSA for different
aerosol size distributions in the North China Plain (NCP) (Tao et al., 2014). As for the aerosol g, it
decreases from 0.667 to 0.602 with the increment of the aerosol RRI. The ambient g values in the NCP
are found within 0.55 and 0.66 (Zhao et al., 2018). Thus, the variations of the RRI have significant
influence on the g. The aerosol optical properties change significantly with the variation of the ambient
aerosol RRI.
The instant DARF values under different RRI are also estimated and the results are illustrated in fig.
11(b). When the aerosol RRI increases from 1.4 to 1.5, the DARF varies from -6.17 to -8.35,
corresponding to 15% variation in DARF. This values are in accordance with the work of Moise et al.
(2015), who estimate that an increment of 12% in the DARF occurs when the RRI varies from 1.4 to
1.5. The DARF can change from -4.9 w/m$^2$ to -10.14 w/m$^2$ when the aerosol RRI increase from 1.34
to 1.56, which corresponding to 40% variation in DARF. Great uncertainties may arise when
estimating the aerosol radiative forcing when using a constant RRI. The RRI should be different under
different aerosol conditions. The real time measured RRI should be used rather than a constant RRI
when estimating the ambient aerosol optical and radiative properties. However, the real-time
measurement of ambient aerosol RRI is not available for most of the conditions. Therefore,
parameterization of the ambient aerosol RRI is necessary.
**4 Conclusions**
The ambient aerosol RRI is a key parameter in determining the aerosol optical properties and
knowledge of it can help constrain the uncertainties in aerosol radiative forcing. In this study, the
ambient aerosol $\widetilde{RRI}$ were measured in East Chinafrom 24[th], May to 18[th], June in 2018. Results show
that the ambient aerosol RRI varies over a wide range between 1.34 and 1.56. The RRI increases slowly
with the increment of the aerosol diameter. The mean aerosol RRI values are 1.425±0.031,
1.435±0.041, 1.47±0.059 for aerosol diameter at 200 nm, 300 nm and 450 nm respectively. Probability
distributions of the RRI show that the RRI is more dispersed with the increment of aerosol dimeter,
which reflect the complexing aging processing of the ambient aerosol. The aerosol optical properties
change significantly and the DARF is estimated to vary by 40% corresponding to the variation of the
measured ambient aerosol RRI. The real-time measured RRI should be used rather than a constant RRI
when estimating the ambient aerosol optical and radiative properties.
We find that the ambient aerosol RRI is highly correlated with the $\rho_{eff}$ rather than the main
chemical compositions of aerosols. There is discrepancy between the measured and parameterized RRI
using the concurrently measured main chemical inorganic compositions of ambient aerosol. This might
be resulted from two reasons. The first one is that the aerosol chemical information used for calculation
is the total aerosol loading    as the aerosol chemical compositions may change significantly among
different size. Another one is that the influence of OM of ambient aerosols is not considered. The RRI
of OM varies significantly for different compositions (Moise et al., 2015).
Despite that the RRI is correlated with the $\rho_{eff}$, parameterization scheme of the ambient aerosol
RRI using $\rho_{eff}$ is not available due to the lack of simultaneously measurement. For the first time, the
$\widetilde{RRI}$ and $\widetilde{\rho_{eff}}$ were measured simultaneously using our designed system. A new parameterization
scheme of the ambient aerosol RRI using the $\rho_{eff}$ is proposed based on the field measurement results.
The measured and parameterized RRI agree well with the correlation coefficient of 0.75 and slope of
0.99. This simple scheme is reliable and ready to be used in the calculation of aerosol optical and
radiative properties. The corresponding measurement results can also be further used in climate model.

**Competing interests.** The authors declare that they have no conflict of interest.
**Data availability.** The data used in this study is available when requesting the authors.
**Author contributions.** GZ, CZ, WZ and SG designed and conducted the experiments; PT, TY and
GZ discussed the results.
**Acknowledgments.** This work is supported by the National Natural Science Foundation of China
(41590872) and National Key R&D Program of China (2016YFC020000:Task 5).

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

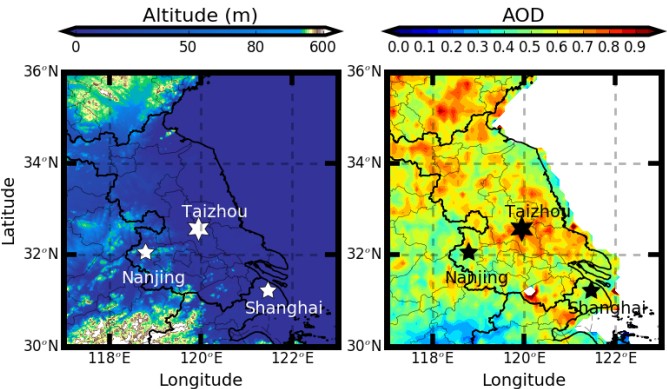


**Figure 1:** Measurement site of Taizhou (marked with stars). Filled colors represent (a) the
topography of the Jianghuai Plain. (b) the average aerosol optical depth at 550nm during the year of
2017 from Moderate Resolution Imaging Spectroradiometer onboard satellite Aqua.


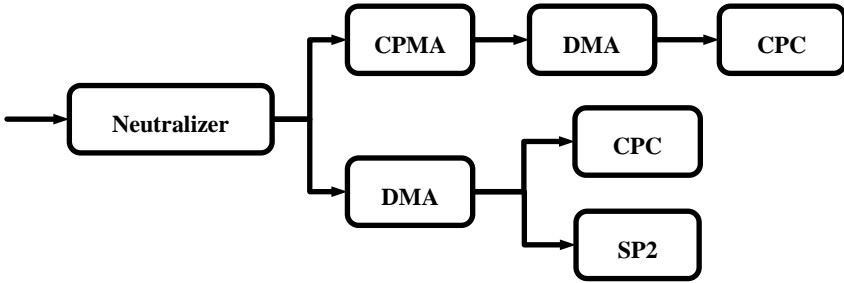

**Figure 2.** Schematic of the instrument setup.



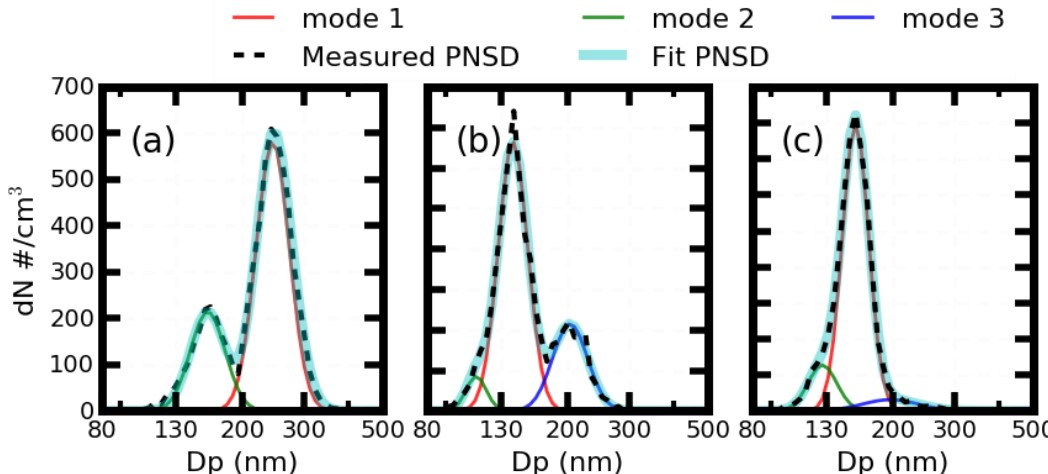


**Figure 3.** The measured aerosol PNSD (black dotted line), fit aerosol number PNSD (blue solid line),
and fit aerosol PNSD at three different mode in different colors that passed through the CPMA. Panel
(a) (b) (c) corresponding to the aerosol mass concentrations of 12, 1 and 1.45 fg respectively.


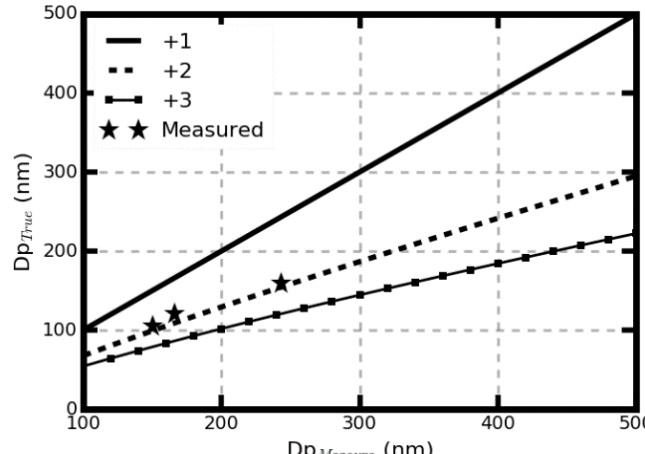


**Figure 4.** The relationship between the measured diameter by the DMA and the calculated aerosol
diameter of different charges in the CPMA-SMPS system.



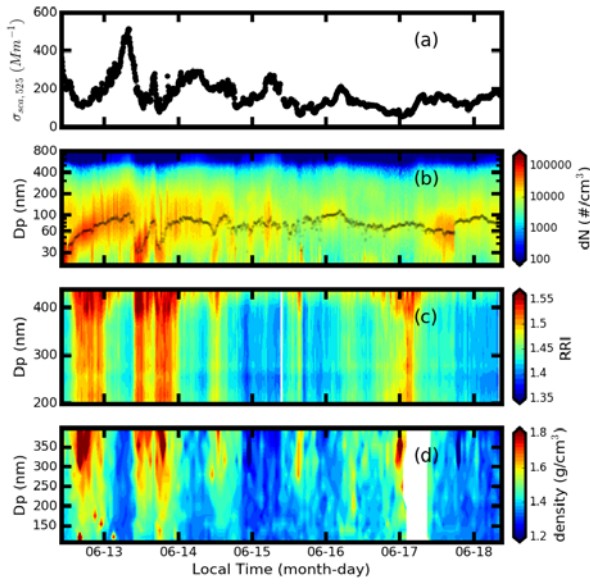


**Figure 5.** Time series of the measured (a) size-resolved RRI in filled color, $\sigma_{sca}$ at 525nm in

black dotted line and (b) the size-resolved $\rho_{eff}$.


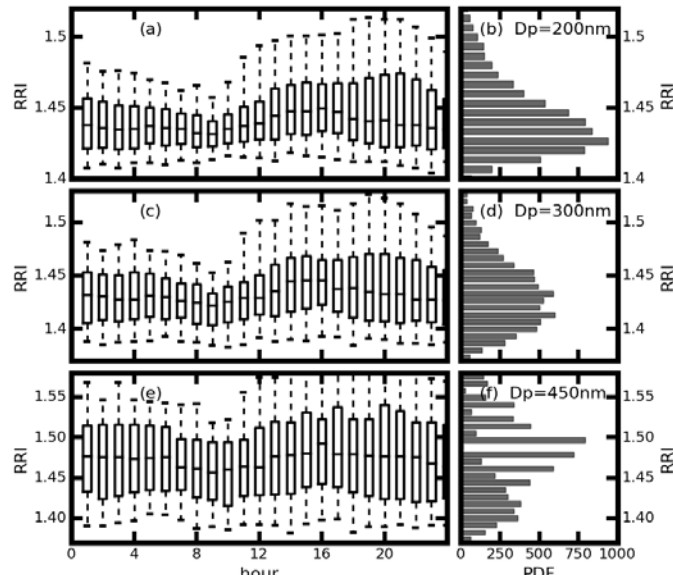


**Figure 6.** Daily variations of the RRI (a), (c) (e), and the probability distribution of the measured
RRI (b), (d) (f) for the (a), (b) 200 nm, (c), (d) 300 nm, and (e), (f) 450nm aerosol respectively. The
box and whisker plots represent the 5th, 25th, 75th and 95th percentiles.

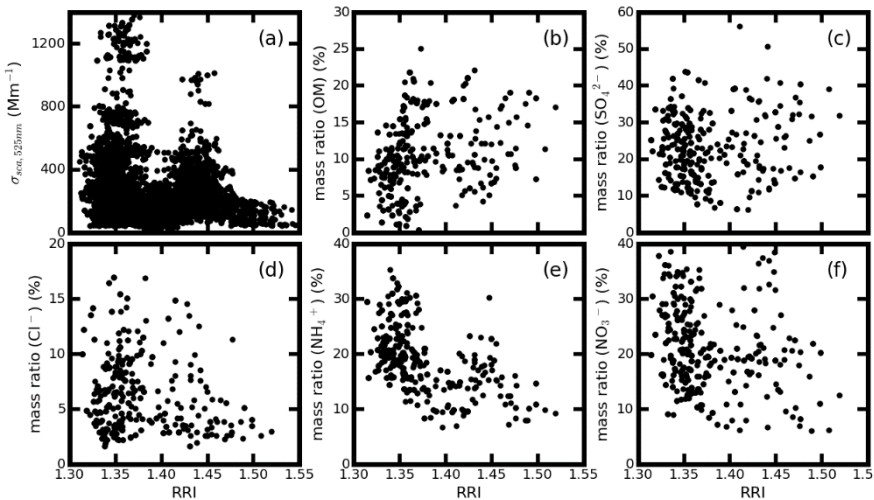


**Figure 7.** Comparison the measured RRI at 300nm with the measured (a) $\sigma_{sca}$ at 525nm, mass

fraction of (b) OM, (c) $SO_4^{2-}$, (d) $Cl^-$, (e) $NH_4^+$ and (f) $NO_3^-$.


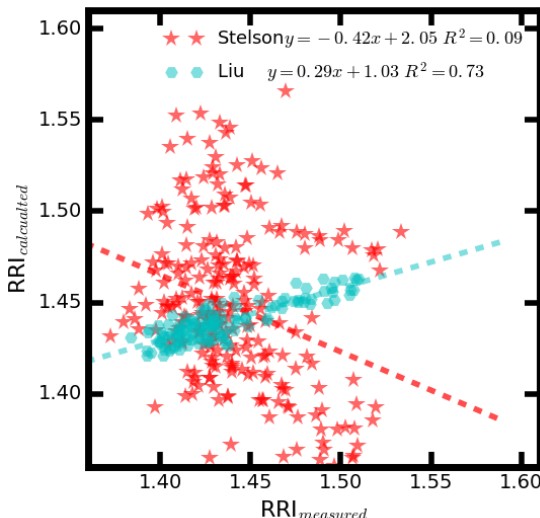


**Figure 8.** Comparison between the measured RRI and calculated RRI using the main aerosol chemical component by applying the method of Stelson (1990) (in red star) and parameterization scheme proposed by Liu and Daum (2008) (in cyan hexagon).



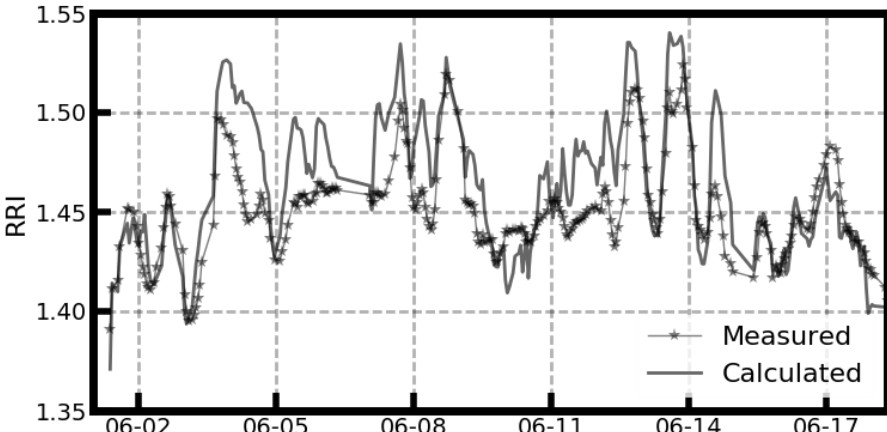


**Figure. 9.** Time series of the measured RRI at 300 nm and the calculated RRI using the aerosol bulk
aerosol optical properties.



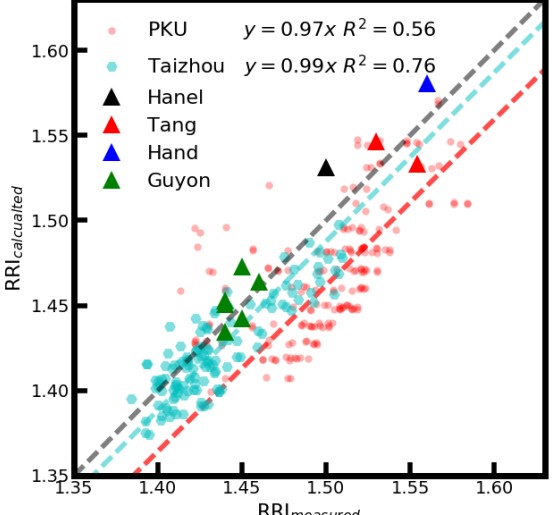


**Figure 10.** Comparison between the measured and calculated RRI for different at PKU (in red circle) and Taizhou (in cyan hexagon) station. The triangle in black , red, blue and green corresponding the data from Hänel (1968), Tang (1996), Hand and Kreidenweis (2002), and Guyon et al. (2003) respectively. The black dashed line is the 1:1 line.



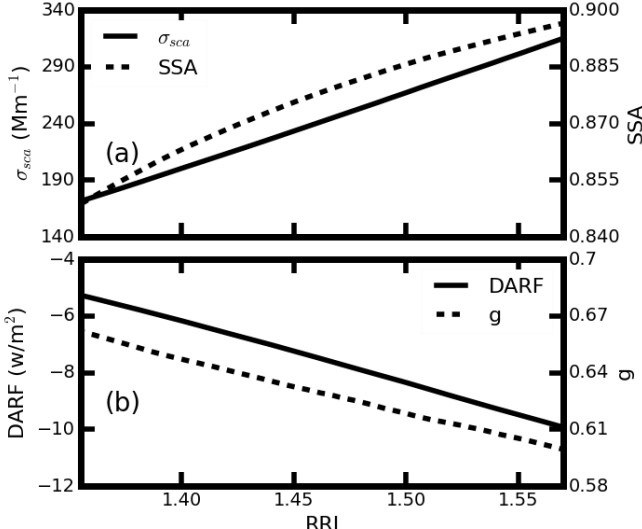


**Figure 11.** Variations of the estimated (a) $\sigma_{sca}$ in solid line, SSA in dotted line, (b) g in dotted

line, and DARF in solid line for different aerosol RRI.
