# Peer review of "A new parameterization scheme of the real part of the ambient urban aerosols refractive index"

_Atmospheric Chemistry and Physics, 2019_

## Referee Comment (RC1) · Anonymous Referee #1 · 24 Apr 2019

General comments: Uncertainty of aerosol optical properties causes further uncertainties in climate prediction in model simulations, in which the real part of the refractive index is important. Thus, determining the aerosol real part of refractive index (RRI) is an important issue. The manuscript entitled "A new parameterization scheme of the real part of the ambient aerosols refractive index" studied the RRI by field measurement in East China. The title is "A new parameterization scheme of the real part of ….", however, as I understood, the parameter scheme is just established by the measurements of the system reported by Zhao et al., (2018b). Moreover, the universality of this parameterization scheme at other location is unknown. Also, the figures and descriptions need be reorganized carefully. Therefore, although this paper focused on the interesting question, it needs further analysis, reorganization, discussion and

clarification to improve the confidence of the results.

Specific comments: 1. Line 26, "reginal" should be "regional". 2. The logics and description of Section "Introduction" are insufficient. 3. I suggest the authors combine some figures, for example, Figure 1, of the supplement into the main of manuscript. 4. Line 153-155, the description of variables in equation (5) is confused. 5. Line 152 and Line 234, all of two equations are denoted as (5). 6. Why not use the vertical profiles of temperature, pressure and water vapor at the times corresponding to the aerosol measurements? 7. Line 234, What's the meaning of  in Equation (5)? 8. Can this method be used at other location and other time? 9. Why do the authors compare a result with other at different time series and measurement site? So, a reliable result should be induced here to evaluate this study. 10. In Section 3.1, what's the relation among the wind speed, T and RH with the $\sigma$scaÅăand mBC? Which should be reflected in descriptions. Otherwise, the results of meteorology measurements are meaningless.

---

## Referee Comment (RC2) · Anonymous Referee #2 · 8 May 2019

General comments:

The real part of the refractive index is surely still uncertain and its impact on the aerosol radiative forcing (ARF) is large. The scope of this manuscript is important. The logic of this manuscript is generally clear, but the following three points should be clarified. Firstly, the title is "A new parameterization scheme of the real part of the ambient aerosols refractive index", so the proposed parameterization must be evaluated in the manuscript, but the evaluation is not enough. The parameterization is based on the measurements at one Chinese site during May-June of the specific year. Generally, the parameterization must be universal, so the proposed one should be tested under various conditions using other measurements at different places and seasons or using a numerical model. Otherwise, I suppose other people do not tend to use the

proposed parameterization. Also, an introduction how to use the parameterization in numerical models, i.e., what is the input and required parameters, may be required. Second, the main conclusion can be led from Figure 4. However, Figure 4 only indicates that Equation (1) is applicable for the effective particle (I understand this is also one of the findings in this study). I expect the clear evidence of the relationship between measured-RRI and calculated-RRI, as shown in Figures S8 and S9. Finally, in the result and discussion of section 3.4, the authors estimated the ARF, but the objectives of this section may be sidetracked. Here, the authors should discuss the impact of the parameterization on the ARF, but the conclusion is "the real-time measured RRI be used rather than a constant RRI when estimating the ambient aerosol optical and radiative properties". This conclusion confuses me. When the proposed parameterization is applied to numerical models, is the real-time measured RRI still required? If so, this parameterization is not attractive to modelers. In addition, the experimental conditions of the ARF calculation is unclear (see the below comment). In overall, the manuscript would be acceptable for publication if these comments can be satisfactorily addressed.

Specific comments:

L23 (and L233): Only correlation coefficient is not enough to evaluate the relation. Please add the other statistical metrics. In abstract, the correlation coefficient is 0.75, but the value is 0.76 in Figure 4. Which is right?

L36: Which wavelengths are used?

L103: Zhao et al. (2018b) seems to be still under discussion. The readers cannot trust the method only from the explanation in this manuscript.

L144-145: RI of BC is set at 1.8+0.54i. Do the authors consider a dependence of RI on wavelength?

L159: Please clarify "parameterization aerosol vertical distributions". This information

is very important to estimate the ARF.

L198-200: The RRI was measured at three different wavelengths (200nm, 300nm and 450nm). Here the measured RR is expressed as "1.34-1.56". Can the measured RRI at different wavelengths be combined? Do the authors consider the difference of RRI among the different wavelengths? In addition, is the focusing wavelength consistent to those proposed by the previous studies?

L204-205: Can the authors explain the mechanism of the relationship between effective density and particle size?

Figure 5: Is the instant value or mean? Which wavelength do the authors calculate? Please clarify them.

Figure S8 and S9: They are very interesting. I strongly recommend they are moved to the main text. Can the authors show the same figures estimated from the current study?

Technical comments"

L34: prat –> part

L46: It is better to add "n: refractive index" to the explanation of Equation (1).

L52: ne –> neff is suitable.

Figure S1 (a), S4, S5: Better to be moved to the main text.

---

## Author Comment (AC1) · 13 Jun 2019

Response to reviewer#1

Thanks for the reviewer's helpful suggestions! The comments are addressed point-by-point and responses are listed below.

**Comment:** General comments: Uncertainty of aerosol optical properties causes further uncertainties in climate prediction in model simulations, in which the real part of the refractive index is important. Thus, determining the aerosol real part of refractive index (RRI) is an important issue.

**Reply:** We thank the anonymous reviewer's comments.

**Comment:** The manuscript entitled "A new parameterization scheme of the real part of the ambient aerosols refractive index" studied the RRI by field measurement in East China. The title is "A new parameterization scheme of the real part of : : :.", however, as I understood, the parameter scheme is just established by the measurements of the system reported by Zhao et al., (2018b). Moreover, the universality of this parameterization scheme at other location is unknown.

**Reply:** Thanks for the comment. The objective of this article is to bring up a novel idea of parameterization scheme of real part of the refractive index (RRI) for ambient aerosol. Traditionally, RRI is parameterized by the measurement of ambient aerosol main inorganic components (Han et al., 2009). The influence of organic compositions is ignored. In this work, we found that the ambient aerosol RRI was highly related with the aerosol effective density ($\rho_{eff}$) rather than the chemical components. Thus, a new parameterization scheme of the RRI using the effective density was proposed.

To validate the universality of this parameterization scheme, we conducted another measurement in the campus of Peking University (PKU) (N39°59′, E116°18′), in China, where the aerosol effective density and real part of the refractive index are measured concurrently at $16^{th}$, December in 2018. The RRI were also calculated using the parameterization scheme, $\frac{RRI^2-1}{RRI^2+2} = 0.18\rho_{eff}$. Comparison of the measured and calculated RRI is shown in fig. R1. Results show that the calculated and measured RRI agree well.

[Figure]

**Fig. R1.** Comparison between the measured and calculated RRI at PKU and Taizhou.

**Comment:** Also, the figures and descriptions need be reorganized carefully. Therefore, although this paper focused on the interesting question, it needs further analysis, reorganization, discussion and clarification to improve the confidence of the results.

**Reply:** We thank the anonymous reviewer's comments and suggestions. We have replotted some figures (1, 2, 5 and 6). We also made some revisions at the introduction and discussion sections in the text.

**Comment:** Specific comments: 1. Line 26, "reginal" should be "regional".

**Reply:** Thanks for the comment and we revised it.

**Comment:** 2. The logics and description of Section "Introduction" are insufficient.

**Reply:** Thanks for the comment. We have rewritten the introduction and added some descriptions about our work.

**Comment:** 3. I suggest the authors combine some figures, for example, Figure 1, of the supplement into the main of manuscript.

**Reply:** Thanks for the comment. Fig. 1 is replotted.

**Comment:** 4. Line 153-155, the description of variables in equation (5) is confused.

**Reply:** Thanks for the comment. We added some descriptions in the text. DARF at the TOA is defined as the difference between radiative flux under aerosol-free conditions and aerosol present conditions:

$$DARF = (f_a \downarrow - f_a \uparrow) - (f_n \downarrow - f_n \uparrow) \ (5),$$

where $f_a \downarrow$ and $f_a \uparrow$ are the downward and upward radiative irradiance with aerosol present conditions respectively; the difference between $f_a \downarrow$ and $f_a \uparrow$

($f_a \downarrow - f_a \uparrow$) is the downward radiative irradiance flux with aerosol present conditions; $f_n \downarrow$ and $f_n \uparrow$ correspond to the downward and upward radiative irradiance values under aerosol free conditions respectively; the difference between $f_n \downarrow$ and $f_n \uparrow$ ($f_n \downarrow - f_n \uparrow$) is the downward radiative irradiance flux for aerosol-free conditions (Kuang et al., 2016).

**Comment:** 5. Line 152 and Line 234, all of two equations are denoted as (5).
**Reply:** Thanks for the comment. We have changed the labels for equations.

**Comment:** 6. Why not use the vertical profiles of temperature, pressure and water vapor at the times corresponding to the aerosol measurements?
**Reply:** Thanks for the comment. When estimating the aerosol DARF using the SBDART model, the profiles of temperature, pressure, water vapor and the aerosol vertical profiles are necessary. DARF would be different for different vertical profiles of temperature, pressure and water vapor. In this study, we focus on the influence of aerosol RRI variation on the variations in DARF. The profiles of aerosol temperature, pressure, water vapor should be hold constant. Therefore, we use the mean result of the measured radiosonde profile during the field.

**Comment:** 7. Line 234, what is the meaning of in Equation (5)?
**Reply:** Thanks for the comment. We have changed the equation into $\mathrm{RRI} = \sqrt{\frac{1+0.36\rho_{eff}}{1-0.18\rho_{eff}}}$, which means that the specific refractive index Re is directly related to aerosol density.

**Comment:** 8. Can this method be used at other location and other time?
**Reply:** Thanks for the comment. We have conducted another measurement in Beijing (N39°59′, E116°18′), China, where the aerosol effective density and real part of the refractive index are measured concurrently at 16th, December in 2018. The relationships of the effective density and real part of refractive index are shown in fig. R1. From fig. R1, the results in Beijing agree well with that of Taizhou.

**Comment:** 9. Why do the authors compare a result with other at different time series and measurement site? So, a reliable result should be induced here to evaluate this study.
**Reply:** Thanks for the comment. We compare the result with that of Liu and Daum (2008) to demonstrate that their parameterization scheme proposed is not applicable in China. The study of Liu and Daum (2008) is currently the only work that have tried to bridge the effective density and real part of refractive index. The effective density and RRI in their work were estimated using the aerosol chemical components but not the in-situ measurements of effective density and RRI. At the same time, the influence of organic aerosols components on aerosol RRI is not considered in their work.

**Comment:** 10. In Section 3.1, what's the relation among the wind speed, T and RH with the scattering coefficient and mBC? Which should be reflected in descriptions. Otherwise, the results of meteorology measurements are meaningless.

**Reply:** Thanks for the comment. The corresponding contents were removed from the text.

Han, Y., Lü, D., Rao, R., Wang, Y. (2009) Determination of the complex refractive indices of aerosol from aerodynamic particle size spectrometer and integrating nephelometer measurements. Applied Optics 48, 4108-4117.

Kuang, Y., Zhao, C.S., Tao, J.C., Bian, Y.X., Ma, N. (2016) Impact of aerosol hygroscopic growth on the direct aerosol radiative effect in summer on North China Plain. Atmospheric Environment 147, 224-233.

Liu, Y., Daum, P.H. (2008) Relationship of refractive index to mass density and self-consistency of mixing rules for multicomponent mixtures like ambient aerosols. Journal of Aerosol Science 39, 974-986.

---

## Author Comment (AC2)

Response to reviewer#2

Thanks for the reviewer's helpful suggestions! The comments are addressed point-by-point and responses are listed below.

**Comment:** General comments: The real part of the refractive index is surely still uncertain and its impact on the aerosol radiative forcing (ARF) is large. The scope of this manuscript is important. The logic of this manuscript is generally clear, but the following three points should be clarified.

**Reply:** Thanks for the comments.

**Comment:** Firstly, the title is "A new parameterization scheme of the real part of the ambient aerosols refractive index", so the proposed parameterization must be evaluated in the manuscript, but the evaluation is not enough. The parameterization is based on the measurements at one Chinese site during May-June of the specific year. Generally, the parameterization must be universal, so the proposed one should be tested under various conditions using other measurements at different places and seasons or using a numerical model. Otherwise, I suppose other people do not tend to use the proposed parameterization

**Reply:** Thanks for the comment. The objective of this article is to bring up a novel idea of parameterization scheme of real part of the refractive index (RRI) for ambient aerosol. Traditionally, RRI is parameterized by the measurement of ambient aerosol main inorganic components (Han et al., 2009). The influence of organic compositions is ignored. In this work, we found that the ambient aerosol RRI was highly related with the aerosol effective density ($\rho_{eff}$) rather than the chemical components. Thus, a new parameterization scheme of the RRI using the effective density was proposed.

To validate the universality of this parameterization scheme, we conducted another measurement in the campus of Peking University (PKU) (N39°59′, E116°18′), in China, where the aerosol effective density and real part of the refractive index are measured concurrently at 16th, December in 2018. The RRI were also calculated using

the parameterization scheme, $\frac{RRI^2-1}{RRI^2+2} = 0.18\rho_{eff}$. Comparison of the measured and calculated RRI is shown in fig. R1. Results show that the calculated and measured RRI show good consistence.

[Figure]

**Fig. R1.** Comparison between the measured and calculated RRI at PKU and Taizhou.

**Comment**: Also, an introduction how to use the parameterization in numerical models, i.e., what is the input and required parameters, may be required.

**Reply:** Our parameterization scheme is simple and easily used in numerical models because the effective density is the only parameter as input. We have demonstrated that the traditional method of calculating the RRI using aerosol main chemical components can have significant bias because the effects of organic aerosol is not considered. We added some discussions in the manuscript correspondingly.

**Comment:** Second, the main conclusion can be led from Figure 4. However, Figure 4 only indicates that Equation (1) is applicable for the effective particle (I understand this is also one of the findings in this study). I expect the clear evidence of the relationship between measured-RRI and calculated-RRI, as shown in Figures S8 and S9.

**Reply:** Thanks for the comment. We have replotted the figure 4.

**Comment:** Finally, in the result and discussion of section 3.4, the authors estimated the ARF, but the objectives of this section may be side tracked. Here, the authors

should discuss the impact of the parameterization on the ARF, but the conclusion is "the real-time measured RRI be used rather than a constant RRI when estimating the ambient aerosol optical and radiative properties". This conclusion confuses me. When the proposed parameterization is applied to numerical models, is the real-time measured RRI still required? If so, this parameterization is not attractive to modelers. In addition, the experimental conditions of the ARF calculation is unclear (see the below comment).

**Reply:** Thanks for the comments. Traditionally, a constant RRI is used when estimating the DARF. As shown in section 3.4, large uncertainties may arise when estimating the DARF using a constant RRI. The real time measured RRI should be used rather than a constant RRI in order to estimate the ambient aerosol optical and radiative properties with high accuracy. However, the real-time measurement of ambient aerosol RRI is not available for most of the conditions. Our proposed parameterization scheme can act as a substitute for real-time RRI.

We added some descriptions of method for ARF calculation in section S3 in the supplementary material.

**Comment:** In overall, the manuscript would be acceptable for publication if these comments can be satisfactorily addressed.

**Reply:** Thanks for the comment.

**Comment:** Specific comments: L23 (and L233): Only correlation coefficient is not enough to evaluate the relation. Please add the other statistical metrics.

**Reply:** Thanks for the comment. We have added the slope in the manuscript.

**Comment:** In abstract, the correlation coefficient is 0.75, but the value is 0.76 in Figure 4. Which is right?

**Reply:** Thanks for the comment. This is a typo and we corrected it.

**Comment:** L36: Which wavelengths are used?

**Reply:** Thanks for the comment. The wavelength range between 0.2 and 5 um is used for calculating the radiative forcing (Marshall et al., 1995; Moise et al., 2015). We have added the description in the text.

**Comment:** L103: Zhao et al. (2018b) seems to be still under discussion. The readers cannot trust the method only from the explanation in this manuscript.

**Reply:** Thanks for the comment. We added some discussions about this method in the manuscript. Before the measurement, this system is calibrated with ammonia sulfate (RRI=1.52). After calibration, ammonium chloride is used to validate the method of deriving the RRI from SP2 for different aerosol diameters. The RRI value of ammonium chloride is 1.642 (Lide, 2006). The retrieved RRI of ammonium chloride is in the range between 1.624 and 1.656. Therefore, this measurement system can measure the ambient aerosol RRI with high accuracy.

**Comment:** L144-145: RI of BC is set at 1.8+0.54i. Do the authors consider a dependence of RI on wavelength?

**Reply:** Thanks for the comment. The RI value of 1.8+0.54i is frequently used in estimating the radiative effects of BC particles (Bond et al., 2013; Zhao et al., 2018). The dependence of RI on wavelength for BC particle is not well studied yet (Bond and Bergstrom, 2006). Therefore, a constant RI of BC at different wavelength is used in estimating the DARF.

**Comment:** L159: Please clarify "parameterization aerosol vertical distributions". This information is very important to estimate the ARF.

**Reply:** Thanks for the comment. We have added some descriptions in the section 3 of the supplementary material to introduce the method of calculating the aerosol vertical profiles.

**Comment:** L198-200: The RRI was measured at three different wavelengths (200nm, 300nm and 450nm). Here the measured RR is expressed as "1.34-1.56". Can the measured RRI at different wavelengths be combined? Do the authors consider the difference of RRI among the different wavelengths? In addition, is the focusing wavelength consistent to those proposed by the previous studies?

**Reply:** Thanks for the comment. The light scattering is measured by SP2 at the wavelength of 1064 and the measured RRI corresponds to the wavelength of 1064 nm. This system is no capable of measuring the RRI among different wavelengths. However, the measured RRI of ambient inorganic aerosols has little variation among different wavelengths. The RRI for $(NH_4)_2SO_4$ varies by 0.02 and less than 0.01 for wavelengths between 400 nm and 700 nm (Cotterell et al., 2017).

We conducted optical closure studies to demonstrate that the measured RRI at 1064 nm is applicable at other wavelength. First, the scattering coefficients ($\sigma_{sca}$) at wavelengths of 450, 525 and 635 nm were calculated using the measured refractive index at 1064 nm and Mie model (Bohren and Huffman, 2007) using the measured aerosol particle number size distribution and the BC mixing states. Then the calculated $\sigma_{sca}$ are compared with the measured $\sigma_{sca}$ by an nephelometer (Aurora 3000, Ecotech, Australia) (Müller et al., 2011). The Aurora 3000 is capable of measuring the $\sigma_{sca}$ at 450, 525 and 635 nm. The scattering truncation and non-Lambertian error was corrected using the same method as that of Ma et al. (2011). The comparison of measured and calculated $\sigma_{sca}$ are shown in fig. R2. The measured and calculated $\sigma_{sca}$ show good consistence, demonstrating the measured RRI using our measurement system is applicable in other wavelength.

[Figure]

**Figure R2.** Comparison between the measured scattering coefficient and calculated scattering coefficient at (a) 450 nm, (b) 525 nm and (c) 635 nm.

**Comment:** L204-205: Can the authors explain the mechanism of the relationship between effective density and particle size?

**Reply:** Thanks for the comment. The difference of the effective density among different particle size should be resulted from the different chemical compositions. Based on the previous measurements of the size-resolved chemical compositions using a MOUDI, the mass fraction of OM decreases with the increment of aerosol diameter (Hu et al., 2012). At the same time, the effective density of OM is lower than the other inorganic compositions. Thus, the effective density increases with the increment of aerosol diameter.

**Comment:** Figure 5: Is the instant value or mean? Which wavelength do the authors calculate? Please clarify them.

**Reply:** Thanks for the comment. The calculated DARF from SBDARF is an instant value. The instant DARF is calculated over the wavelength range between 0.25 μm and 4 μm. We have added the descriptions in the text.

**Comment:** Figure S8 and S9: They are very interesting. I strongly recommend they are moved to the main text. Can the authors show the same figures estimated from the current study?

**Reply:** Thanks for the comment. Fig. 4 is replotted. Fig. S8 and fig. S9 were merged into figure 5.

**Comment:** Technical comments" L34: prat –> part

**Reply:** Thanks for the comment. We have revised it.

**Comment:** L46: It is better to add "n: refractive index" to the explanation of Equation (1).

**Reply:** Thanks for the comment. We have changed the equation as

$$RRI_{eff} = \sum_i (f_i \cdot RRI_i)$$

Where $f_i$ and $RRI_i$ are the volume fraction and real part of refractive index of known composition *i*.

**Comment:** L52: ne –> neff is suitable.

**Reply:** Thanks for the comment. We changed the "n" into $RRI_{eff}$ in the manuscript.

**Comment:** Figure S1 (a), S4, S5: Better to be moved to the main text.

**Reply:** Thanks for the comment. Fig. S1 is moved into Fig. 2 in the text and part of fig. S4 is moved to the main text.

Bohren, C.F., Huffman, D.R., (2007) Absorption and Scattering by a Sphere, Absorption and Scattering of Light by Small Particles. Wiley-VCH Verlag GmbH, pp. 82-129.

Bond, T.C., Bergstrom, R.W. (2006) Light Absorption by Carbonaceous Particles: An Investigative Review. Aerosol Science And Technology 40, 27-67.

Bond, T.C., Doherty, S.J., Fahey, D.W., Forster, P.M., Berntsen, T., DeAngelo, B.J., Flanner, M.G., Ghan, S., Karcher, B., Koch, D., Kinne, S., Kondo, Y., Quinn, P.K., Sarofim, M.C., Schultz, M.G., Schulz, M., Venkataraman, C., Zhang, H., Zhang, S., Bellouin, N., Guttikunda, S.K., Hopke, P.K., Jacobson, M.Z., Kaiser, J.W., Klimont, Z., Lohmann, U., Schwarz, J.P., Shindell, D., Storelvmo, T., Warren, S.G., Zender, C.S. (2013) Bounding the role of black carbon in the climate system: A scientific assessment. Journal Of Geophysical Research-Atmospheres 118, 5380-5552.

Cotterell, M.I., Willoughby, R.E., Bzdek, B.R., Orr-Ewing, A.J., Reid, J.P. (2017) A complete parameterisation of the relative humidity and wavelength dependence of the refractive index of hygroscopic inorganic aerosol particles. Atmospheric Chemistry and Physics 17, 9837-9851.

Han, Y., Lü, D., Rao, R., Wang, Y. (2009) Determination of the complex refractive indices of aerosol from aerodynamic particle size spectrometer and integrating nephelometer measurements. Applied Optics 48, 4108-4117.

Hu, M., Peng, J., Sun, K., Yue, D., Guo, S., Wiedensohler, A., Wu, Z. (2012) Estimation of size-resolved ambient particle density based on the measurement of aerosol number, mass, and chemical size distributions in the winter in Beijing. Environ Sci Technol 46, 9941-9947.

Lide, D.R. (2006) Handbook of Chemistry and Physics, 86th Edition Edited(National Institute of Standards and Technology). Journal of the American Chemical Society 128, 5585-5585.

Ma, N., Zhao, C.S., Nowak, A., Müller, T., Pfeifer, S., Cheng, Y.F., Deng, Z.Z., Liu, P.F., Xu, W.Y., Ran, L., Yan, P., Göbel, T., Hallbauer, E., Mildenberger, K., Henning, S., Yu, J., Chen, L.L., Zhou, X.J., Stratmann, F., Wiedensohler, A. (2011) Aerosol optical properties in the North China Plain during HaChi campaign: an in-situ optical closure study. Atmos. Chem. Phys. 11, 5959-5973.

Marshall, S.F., Covert, D.S., Charlson, R.J. (1995) Relationship between asymmetry parameter and hemispheric backscatter ratio: implications for climate forcing by aerosols. Applied Optics 34, 6306-6311.

Moise, T., Flores, J.M., Rudich, Y. (2015) Optical properties of secondary organic aerosols and their changes by chemical processes. Chemical Reviews 115, 4400-4439.

Müller, T., Laborde, M., Kassell, G., Wiedensohler, A. (2011) Design and performance of a three-wavelength LED-based total scatter and backscatter integrating nephelometer. Atmos. Meas. Tech. 4, 1291-1303.

Zhao, G., Zhao, C., Kuang, Y., Bian, Y., Tao, J., Shen, C., Yu, Y. (2018) Calculating the aerosol asymmetry factor based on measurements from the humidified nephelometer system. Atmospheric Chemistry and Physics 18, 9049-9060.

---

## Referee Report (RR1)

**Comments on "A new parameterization scheme of the real part of the ambient aerosols refractive index" by Zhao et al.**

This study deals with a very interesting and important issue, i.e., parameterization of aerosol refractive index, which is essential for the estimation of aerosol direct radiative forcing. I read the manuscript and the authors' response with great interest. However, after careful evaluation, I agree with the other reviewer that this study is not suitable for publication in ACP as "it needs further analysis, reorganization, discussion and clarification to (prove) improve the confidence of the results (reviewer 1)". I will expand a bit on these issues as detailed below.

**Major Comments:**

The parameterization scheme of the current study ($Re = (RRI^2-1)/(RRI^2+2) = 0.18\rho_{eff}$) is in principle a justification / an update of the scheme proposed by Liu and Daum (2008) ($Re = (RRI^2-1)/(RRI^2+2) = 0.23\rho_{eff}^{0.39}$) based on a new dataset measured at Taizhou in China for 7 days in June 2018. The major concern from the other reviewers is whether the new parameterization is universal and applicable in global and climate models as suggested by the authors. I am not convinced by the author's arguments because of the following reasons.

1. Is the new scheme universal and better than the one from Liu and Daum (2008)?

   If a scheme is universal, it should not only explain one dataset, but also be applicable and compatible for other datasets. To come up with their parameterization scheme, Liu and Daum (2008) studied the relationship of refractive index to mass density (index-density relationship) for over 4000 pure materials and for aerosol particles. Note that, in Liu and Daum (2008), the summarized pure materials include organics, and investigated aerosol data cover aerosol samples from Amazon (Guyon et al., 2003), which is expected to constitute significant fraction of organics. Thus, it is not appropriate for the authors to make a statement "the influence of organic aerosols components on aerosol RRI is not considered in their work (L270-271)".

   In Fig. C1, I compared the results of the current study with Liu and Daum (2008) by overlaying the data from the Taizhou site (small light blue dots) and from the PKU site (small pink dots) onto the original Fig. 3 of Liu and Daum (2008). Interestingly, the new datasets are not much different from the ones already summarized by Liu and Daum (2008), as they fit well into the data clouds within the same mass density range. For the whole data population, it appears that Liu and Daum's scheme (black solid line) is still the best approach to describe the overall index-density relationship. Because the new parametrization from the current study (blue solid line) is not able to represent the general trend in the existing dataset (over 4000 pure materials marked by small grey dots), especially it failed to explain the aerosol data from early field campaign/laboratory studies (marked by big black triangles).

   Depending on to which degree one would like the schemes to represent the variability, for the Taizhou site the predicted average RRI (~1.44) by Liu and Daum's scheme is in a reasonable agreement with the observed 28-day average RRI of 1.425, 1.435 and

1.47 for 200 nm, 300 nm and 450 nm particles, respectively, which is probably already good enough for global and climate model applications.

On the other hand, when speaking of explaining the detailed temporal and spatial variability, the prediction of the new parameterization at the PKU site is quite scattered with y = 1.0x and $R^2$ = 0.03 (see my Major Comments 2). For example, a prediction of RRI ~1.5 with the new parameterization scheme at the PKU site may correspond to a variability of real/observed RRI from 1.42 to 1.58 (Fig. C2-A).

Thus, it is unlikely that the new parameterization scheme from the current study is universal and applicable to global and climate models. In my opinion, the intrinsic scattering of the index-density relationship (Fig. C1) implies that a perfect parameterization may not be even possible. If a compromise has to be made, Liu and Daum's scheme still seems to be optimal choice in terms of universality.

[Figure]

**Fig. C1.** Dependence of the effective refractive index (Re = (RRI²-1)/(RRI²+2)) on the effective mass density ($\rho_{eff}$) for pure materials and for aerosol particles. The new data sets from the current study (small light blue dots for Taizhou and small pink dots for PKU) overlay the original Fig. 3 of Liu and Daum (2008). The data summarized by Liu Daum (2008) includes over 4000 pure materials (small grey dots, including organics, inorganics and minerals, www.knovel.com), as well as ambient aerosol and lab generated surrogate with chemical compositions representing ambient aerosols (big black triangles) (Hänel, 1968; Tang and Munkelwitz, 1994; Hand and Kreidenweis, 2002; Guyon et al. 2003).

2. Consistence between the PKU and Taizhou sites?

When comparing the consistency of different dataset, I find that I cannot reproduce the results of Fig. 6 in the revised manuscript (also shown here as Fig. C2-B). While the authors provided a $R^2$ of 0.47 for the PKU site (Fig. C2-B), using the same fit function (y = ax by forcing intercept = 0), I received a coefficient of determination ($R^2$) of only 0.03 for the same dataset (Fig. C2-A), not sufficient to support the authors' argument about consistence between the PKU and Taizhou sites. Apparently, the authors have selected the "good" slope (1.0) of y = ax and the "better" $R^2$ (0.47) of y = ax + b to justify the advantage of their method in Fig. 6 and relevant text. This is misleading. Such a way of selectively presenting results is a serious issue and has to be corrected.

[Figure]

**Fig. C2.** Comparison of measured and predicted RRI by the parameterization scheme of Zhao et al. (ACPD). **A**, my re-calculation with data from Fig. 6 of Zhao et al. (ACPD). **B**, the original Fig. 6 of Zhao et al. (ACPD).

**Other Comments:**

1. Abstract: The retrieved RRI is for pure scattering aerosols (or may be extended for the coating materials when calculating the effective refractive index of mixed black particles?), while the effective density is measured for all aerosols (both scattering and absorbing aerosols). Direct comparison between the two measured quantifies may induce uncertainties, and should be justified or at least clarified.

2. Abstract and section 3.2: I suggest to remove "rather than the main chemical components" from "We find that the ambient aerosol RRI is highly related with the aerosol effective density ($\rho_{eff}$) rather than the main chemical components", or change 'related' to 'correlated'. This is because both refractive index and effective density are determined by main chemical components of aerosol particles. Even for the proposed application in global or climate model (calculation of RRI from $\rho_{eff}$), one would still need the simulated chemical compositions to calculate $\rho_{eff}$ (see my Other Comments 3).

   Along this line, section 3.2 should be substantially revised by including more detailed and thorough evidences and discussions. See my concerns below.

   The major argument/results presented in section 3.2 to support the conclusion "the ambient aerosol RRI is highly related with the aerosol effective density ($\rho_{eff}$) rather than the main chemical components" are Fig. 4, Fig. 5 and Fig. S6 (in the revised manuscript). However, RRI and chemical compositions in these comparisons were not taken from the same aerosol group. The RRI were taken from aerosol of a certain size (i.e., 300 nm) while the chemical compositions were taken from $PM_{2.5}$ either for direct comparison (Fig. 4 and Fig. S6) or for calculating the RRI (Fig. 5) (The water-soluble ions were from $PM_{2.5}$; whether ECOC measurements were from $PM_{2.5}$ or $PM_{10}$ are not clear in the text of section 2.1). In this case, even the same parameter may differ from each other. For example, one can clearly see multiple modes in the comparison of main aerosol components and RRI in Fig. S6.

Such comparison can be misleading because there is a danger that the readers might get an impression that the commonly used mixing rules in calculating refractive index wouldn't work for ambient aerosols, e.g., volume linear mixing rule, Maxwell-Garnet and Brüggemann mixing rule, partial molar refraction mixing rule, Lorentz-Lorenz mixing rule etc. (those requires information of chemical compositions in a mixed system). A direct consequence has already been shown in Fig. 5, where the authors delivered a message that Stelson's approach (Stelson, 1990) of calculating refractive index with partial molar refraction mixing rule did not work for the Taizhou case. This is unfair because the mismatch of different aerosol population in this comparison may to some (large) extend lead to the very scattered data points of the Stelson's approach in Fig. 5.

3. The authors' response about "how to use the parameterization in numerical models, i.e., what is the input and required parameters, may be required" is not adequate. Modelers understand "the effective density is the only parameter as input", but the real question is how to determine the effective density in the model, which hasn't been answered. I guess that one would still need to calculate $\rho_{eff}$ from densities of individual simulated chemical composition. This procedure however, may be hampered by the lack of density information of organic carbons and mixing state of black carbon, etc.

4. Concerning Reviewer 2's comment "Zhao et al. (2018b) seems to be still under discussion. The readers cannot trust the method only from the explanation in this manuscript.", the authors may want to refer to the work of Zhang et al. (JGR, 2018), where the method of combining DMA and SP2 to retrieve the real part of the refractive index of pure scattering aerosol particles has been proposed and published.

5. L50: "main aerosol" is duplicated.

6. L198-199: could it be that statistics of RRI at 200 and 300 nm is better that at 450 nm? Because the scattering signal of SP2 may become saturated for a large fraction of particles at 450 nm, and reduce the sample size. How were the double charged particles treated?

7. L147-151: "SSA is defined as the ratio of $\sigma_{sca}$ to $\sigma_{ext}$, which reflects concentration of the absorbing aerosol (Tao et al., 2014) to some extent. The $g$ expresses the distribution of the scattering light intensity in different directions (Zhao et al., 2018a). The $\sigma_{ext}$, SSA and $g$ are the most important three factors that influence the aerosol radiative properties in radiative calculation (Kuang et al., 2015)." L169-170: "… the difference between fn↓ and fn↑ (fn↓ - fn↑) is the downward radiative irradiance flux for aerosol-free conditions (Kuang et al., 2016)."

It does not seem to be appropriate to cite these references here, because such statements are rather classical textbook knowledge.

8. L192: change "The RRI and $\rho_{eff}$ vary…" to "The RRI varies".

9. L201-202: change "at about 15:00 in the morning" to "at about 15:00 in the afternoon" and change "at around 9:00 in the afternoon" to "at around 9:00 in the morning".

10. L214-215: "…Thus, the effective tend to increase with the increment of aerosol diameter." sounds like a broken sentence.

11. L254: "one day" instead of "one days".

12. Fig. 5: The caption is confusing and needs to be revised. "Comparison between the measured RRI and calculated RRI using the main aerosol chemical component from Stelson (1990) (in red star)…" Do the authors mean using the same aerosol chemical species those are needed for applying the Stelson's method, but the concentrations of the chemical components are still the measured ones from this study? If yes, please revise the caption. This is related to reviewer's comment about "Why do the authors compare a result with other at different time series and measurement site?"

**Reference**

Y. Liu, P. H. Daum, Relationship of refractive index to mass density and self-consistency of mixing rules for multicomponent mixtures like ambient aerosols. Journal of Aerosol Science 39, 974-986 (2008).

G. HÄNEL, The real part of the mean complex refractive index and the mean density of samples of atmospheric aerosol particles. Tellus 20, 371-379 (1968).

J. L. Hand, S. M. Kreidenweis, A New Method for Retrieving Particle Refractive Index and Effective Density from Aerosol Size Distribution Data. Aerosol Science and Technology 36, 1012-1026 (2002).

N. Tang, H. R. Munkelwitz, Water activities, densities and refractive indices of aqueous sulfates and sodium nitrate droplets of atmospheric importance. J. Geophys. Res. 99, 18801-18808 (1994).

P. Guyon et al., Refractive index of aerosol particles over the Amazon tropical forest during LBA-EUSTACH 1999. J. Aerosol Sci. 34, 883-907 (2003).

Y. Zhang et al., Sizing of Ambient Particles from a Single-Particle Soot Photometer Measurement to Retrieve Mixing State of Black Carbon at a Regional Site of the North China Plain. Journal of Geophysical Research: Atmospheres 123, 12,778-712,795 (2018).

A. W. Stelson, Urban aerosol refractive index prediction by partial molar refraction approach, ES&T, 24, 1990.

---

## Referee Report (RR2)

This study designed a field measurement system and found a new method to better calculate the real part of refractive index where information aerosol density is available. The topic of the study is undoubtedly of high scientific and practical importance. On the whole, the experimental methodology and data analyzing procedures look to be correct and the findings will significantly improve the estimation of aerosol radiative forcing. Therefore, this manuscript is helpful for the audience of atmospheric chemistry and physics, but not without a major revision. Some comments and suggestions are listed below.

**Main Points**

(1) I am a little concerned about the title "A new parameterization scheme of …". For a parameterization scheme, parameters should be changeable at different situation. In this study, a coefficient of 0.18 is obtained from two field measurements and should be applicable to the polluted regions. But as shown in equation 2, the definition of this coefficient suggests that it would vary with the molecular polarizability and molecular weight of the aerosols composition. As a result, I am not sure whether the coefficient 0.18 would be applicable for sea salt aerosols or organic-dominated aerosols. An alternative way is to limit the scope to a certain kind of atmospheric condition. By the way, is the parameter derived from the two measurements exactly the same?

(2) The usage of "RRI" or its related form in the paper is always confusing. Whether it is measured or calculated, whether it is size-resolved or not. It is suggested to double-check the usage of "RRI" or its related form all through the paper. A clear parameterization table with definition would be helpful for readership to better understand the paper. Also, why the authors use different size-resolved RRI at different places? I think there are size-resolved RRI at 200nm, 250nm and 300nm in different discussion.

(3) The structure of the current manuscript is not well organized. For example, in the Data and Methods part, the readers will have an expression that this paper is based on one campaign in Taizhou. However, in the discussion part (line 298), the measurement data at PKU site is also used, but without any description of the measurement. Another example is that there is too much background discussion and methodology in the conclusion part.

(4) The authors may need to be more careful on some statements made in the manuscript. For example, line 344, "Our proposed parameterizations scheme is a perfect substitute" is not appropriate for a scientific paper. Also, line 258 "the RRI tend to increase with the OM mass fraction ratio". I don't recognize clear trend in fig 7. A simple hypothesis testing may be needed here.

(5) The mode 1, 2, 3 derived from DMA-CPMA-CPC measurement are considered as light absorbing aerosols, scattering aerosols and double charged aerosols. Though the aerosols with lower density are very likely the fresh emitted light absorbing aerosols, those with higher density could also be fully aged light absorbing aerosols. In my opinion, mode 1 is more like "fractal aerosols" and mode 2 is "compact aerosols". This definition may not influence the final conclusion, but still need to be carefully discussed. One suggestion is to compare the aerosol number in Mode 1 and BC number

concentration measured by SP2 at different size to make sure they are comparable.

(6) Line 20, the authors stated "For the first time, the size-resolved ambient aerosol RRI and ρeff are measured simultaneously by our designed measurement system". Since the particle size (also chemical compositions) is linked to distinct formation processes and stages of haze development, such as nucleation and growth from clean, transition, to polluted periods (Guo et al., Elucidating severe urban haze formation in China, *Proc. Natl. Acad. Sci. USA* **111**, 17373, 2014; Wang et al., Persistent sulfate formation from London Fog to Chinese Haze, *Proc. Natl. Acad. Sci. USA* **113**, 13630, 2016), it would be necessary that a connection between the RRI and haze development is identified.

(6) I also believe that some references in this paper were outdated, and a significant effort is needed to address such. Below are some examples.

Line 26, The author stated that "Atmospheric aerosols can significantly influence the regional air quality and climate system by scattering and absorbing the solar radiation (Seinfeld et al., 1998)". Several other most recent papers on this topic need to be discussed (i.e., An et al., Severe haze in Northern China: A synergy of anthropogenic emissions and atmospheric processes, *Proc. Natl. Acad. Sci. USA* **116**, 8657, 2019; Zhang et al., Formation of urban fine particulate matter, *Chem. Rev.* **115**, 3803, 2015; Wang et al., Light absorbing aerosols and their atmospheric impacts, *Atmos. Environ.* **81**, 713, 2013).

**Technical comments**

Line 13, delete "Mainly"
Line 15, change "Results" to "The results"
Line 16, the sentence "vary by 40% corresponding to the variation of the measured aerosol RRI" is confusing.
Line 19, delete "schemes"
Line 173, "relations ship" should be "relationship"
Line 301, "equation 7" should be the equation 9?

---

## Editor Decision (ED1)

General comments: Uncertainty of aerosol optical properties causes further uncertainties in climate prediction in model simulations, in which the real part of the refractive index is important. Thus, determining the aerosol real part of refractive index (RRI) is an important issue. The manuscript entitled "A new parameterization scheme of the real part of the ambient aerosols refractive index" studied the RRI by field measurement in East China. The title is "A new parameterization scheme of the real part of . . ..", however, as I understood, the parameter scheme is just established by the measurements of the system reported by Zhao et al., (2018b). Moreover, the universality of this parameterization scheme at other location is unknown. Also, the figures and descriptions need be reorganized carefully. Therefore, although this paper focused on the interesting question, it needs further analysis, reorganization, discussion and

clarification to improve the confidence of the results.

Specific comments: 1. Line 26, "reginal" should be "regional". 2. The logics and description of Section "Introduction" are insufficient. 3. I suggest the authors combine some figures, for example, Figure 1, of the supplement into the main of manuscript. 4. Line 153-155, the description of variables in equation (5) is confused. 5. Line 152 and Line 234, all of two equations are denoted as (5). 6. Why not use the vertical profiles of temperature, pressure and water vapor at the times corresponding to the aerosol measurements? 7. Line 234, What's the meaning of  in Equation (5)? 8. Can this method be used at other location and other time? 9. Why do the authors compare a result with other at different time series and measurement site? So, a reliable result should be induced here to evaluate this study. 10. In Section 3.1, what's the relation among the wind speed, T and RH with the $\sigma$scaÂǎand mBC? Which should be reflected in descriptions. Otherwise, the results of meteorology measurements are meaningless.

[Figure]

The real part of the refractive index is surely still uncertain and its impact on the aerosol radiative forcing (ARF) is large. The scope of this manuscript is important. The logic of this manuscript is generally clear, but the following three points should be clarified. Firstly, the title is "A new parameterization scheme of the real part of the ambient aerosols refractive index", so the proposed parameterization must be evaluated in the manuscript, but the evaluation is not enough. The parameterization is based on the measurements at one Chinese site during May-June of the specific year. Generally, the parameterization must be universal, so the proposed one should be tested under various conditions using other measurements at different places and seasons or using a numerical model. Otherwise, I suppose other people do not tend to use the

proposed parameterization. Also, an introduction how to use the parameterization in numerical models, i.e., what is the input and required parameters, may be required. Second, the main conclusion can be led from Figure 4. However, Figure 4 only indicates that Equation (1) is applicable for the effective particle (I understand this is also one of the findings in this study). I expect the clear evidence of the relationship between measured-RRI and calculated-RRI, as shown in Figures S8 and S9. Finally, in the result and discussion of section 3.4, the authors estimated the ARF, but the objectives of this section may be sidetracked. Here, the authors should discuss the impact of the parameterization on the ARF, but the conclusion is "the real-time measured RRI be used rather than a constant RRI when estimating the ambient aerosol optical and radiative properties". This conclusion confuses me. When the proposed parameterization is applied to numerical models, is the real-time measured RRI still required? If so, this parameterization is not attractive to modelers. In addition, the experimental conditions of the ARF calculation is unclear (see the below comment). In overall, the manuscript would be acceptable for publication if these comments can be satisfactorily addressed.

Specific comments:

L23 (and L233): Only correlation coefficient is not enough to evaluate the relation. Please add the other statistical metrics. In abstract, the correlation coefficient is 0.75, but the value is 0.76 in Figure 4. Which is right?

L36: Which wavelengths are used?

L103: Zhao et al. (2018b) seems to be still under discussion. The readers cannot trust the method only from the explanation in this manuscript.

L144-145: RI of BC is set at 1.8+0.54i. Do the authors consider a dependence of RI on wavelength?

L159: Please clarify "parameterization aerosol vertical distributions". This information

is very important to estimate the ARF.

L198-200: The RRI was measured at three different wavelengths (200nm, 300nm and 450nm). Here the measured RR is expressed as "1.34-1.56". Can the measured RRI at different wavelengths be combined? Do the authors consider the difference of RRI among the different wavelengths? In addition, is the focusing wavelength consistent to those proposed by the previous studies?

L204-205: Can the authors explain the mechanism of the relationship between effective density and particle size?

Figure 5: Is the instant value or mean? Which wavelength do the authors calculate? Please clarify them.

Figure S8 and S9: They are very interesting. I strongly recommend they are moved to the main text. Can the authors show the same figures estimated from the current study?

Technical comments"

L34: prat –> part

L46: It is better to add "n: refractive index" to the explanation of Equation (1).

L52: ne –> neff is suitable.

Figure S1 (a), S4, S5: Better to be moved to the main text.
* * *
**Comments on "A new parameterization scheme of the real part of the ambient aerosols refractive index" by Zhao et al.**

This study deals with a very interesting and important issue, i.e., parameterization of aerosol refractive index, which is essential for the estimation of aerosol direct radiative forcing. I read the manuscript and the authors' response with great interest. However, after careful evaluation, I agree with the other reviewer that this study is not suitable for publication in ACP as "it needs further analysis, reorganization, discussion and clarification to (prove) improve the confidence of the results (reviewer 1)". I will expand a bit on these issues as detailed below.

**Major Comments:**

The parameterization scheme of the current study ($Re = (RRI^2-1)/(RRI^2+2) = 0.18\rho_{eff}$) is in principle a justification / an update of the scheme proposed by Liu and Daum (2008) ($Re = (RRI^2-1)/(RRI^2+2) = 0.23\rho_{eff}^{0.39}$) based on a new dataset measured at Taizhou in China for 7 days in June 2018. The major concern from the other reviewers is whether the new parameterization is universal and applicable in global and climate models as suggested by the authors. I am not convinced by the author's arguments because of the following reasons.

1. Is the new scheme universal and better than the one from Liu and Daum (2008)?

   If a scheme is universal, it should not only explain one dataset, but also be applicable and compatible for other datasets. To come up with their parameterization scheme, Liu and Daum (2008) studied the relationship of refractive index to mass density (index-density relationship) for over 4000 pure materials and for aerosol particles. Note that, in Liu and Daum (2008), the summarized pure materials include organics, and investigated aerosol data cover aerosol samples from Amazon (Guyon et al., 2003), which is expected to constitute significant fraction of organics. Thus, it is not appropriate for the authors to make a statement "the influence of organic aerosols components on aerosol RRI is not considered in their work (L270-271)".

   In Fig. C1, I compared the results of the current study with Liu and Daum (2008) by overlaying the data from the Taizhou site (small light blue dots) and from the PKU site (small pink dots) onto the original Fig. 3 of Liu and Daum (2008). Interestingly, the new datasets are not much different from the ones already summarized by Liu and Daum (2008), as they fit well into the data clouds within the same mass density range. For the whole data population, it appears that Liu and Daum's scheme (black solid line) is still the best approach to describe the overall index-density relationship. Because the new parametrization from the current study (blue solid line) is not able to represent the general trend in the existing dataset (over 4000 pure materials marked by small grey dots), especially it failed to explain the aerosol data from early field campaign/laboratory studies (marked by big black triangles).

   Depending on to which degree one would like the schemes to represent the variability, for the Taizhou site the predicted average RRI (~1.44) by Liu and Daum's scheme is in a reasonable agreement with the observed 28-day average RRI of 1.425, 1.435 and

1.47 for 200 nm, 300 nm and 450 nm particles, respectively, which is probably already good enough for global and climate model applications.

On the other hand, when speaking of explaining the detailed temporal and spatial variability, the prediction of the new parameterization at the PKU site is quite scattered with $y = 1.0x$ and $R^2 = 0.03$ (see my Major Comments 2). For example, a prediction of RRI ~1.5 with the new parameterization scheme at the PKU site may correspond to a variability of real/observed RRI from 1.42 to 1.58 (Fig. C2-A).

Thus, it is unlikely that the new parameterization scheme from the current study is universal and applicable to global and climate models. In my opinion, the intrinsic scattering of the index-density relationship (Fig. C1) implies that a perfect parameterization may not be even possible. If a compromise has to be made, Liu and Daum's scheme still seems to be optimal choice in terms of universality.

[Figure]

**Fig. C1.** Dependence of the effective refractive index ($R_e = (RRI^2-1)/(RRI^2+2)$) on the effective mass density ($\rho_{eff}$) for pure materials and for aerosol particles. The new data sets from the current study (small light blue dots for Taizhou and small pink dots for PKU) overlay the original Fig. 3 of Liu and Daum (2008). The data summarized by Liu Daum (2008) includes over 4000 pure materials (small grey dots, including organics, inorganics and minerals, www.knovel.com), as well as ambient aerosol and lab generated surrogate with chemical compositions representing ambient aerosols (big black triangles) (Hänel, 1968; Tang and Munkelwitz, 1994; Hand and Kreidenweis, 2002; Guyon et al. 2003).

2. Consistence between the PKU and Taizhou sites?

When comparing the consistency of different dataset, I find that I cannot reproduce the results of Fig. 6 in the revised manuscript (also shown here as Fig. C2-B). While the authors provided a $R^2$ of 0.47 for the PKU site (Fig. C2-B), using the same fit function ($y = ax$ by forcing intercept = 0), I received a coefficient of determination ($R^2$) of only 0.03 for the same dataset (Fig. C2-A), not sufficient to support the authors' argument about consistence between the PKU and Taizhou sites. Apparently, the authors have selected the "good" slope (1.0) of $y = ax$ and the "better" $R^2$ (0.47) of $y = ax + b$ to justify the advantage of their method in Fig. 6 and relevant text. This is misleading. Such a way of selectively presenting results is a serious issue and has to be corrected.

[Figure]

**Fig. C2.** Comparison of measured and predicted RRI by the parameterization scheme of Zhao et al. (ACPD). **A**, my re-calculation with data from Fig. 6 of Zhao et al. (ACPD). **B**, the original Fig. 6 of Zhao et al. (ACPD).

**Other Comments:**

1. Abstract: The retrieved RRI is for pure scattering aerosols (or may be extended for the coating materials when calculating the effective refractive index of mixed black particles?), while the effective density is measured for all aerosols (both scattering and absorbing aerosols). Direct comparison between the two measured quantifies may induce uncertainties, and should be justified or at least clarified.

2. Abstract and section 3.2: I suggest to remove "rather than the main chemical components" from "We find that the ambient aerosol RRI is highly related with the aerosol effective density ($\rho_{eff}$) rather than the main chemical components", or change 'related' to 'correlated'. This is because both refractive index and effective density are determined by main chemical components of aerosol particles. Even for the proposed application in global or climate model (calculation of RRI from $\rho_{eff}$), one would still need the simulated chemical compositions to calculate $\rho_{eff}$ (see my Other Comments 3).

   Along this line, section 3.2 should be substantially revised by including more detailed and thorough evidences and discussions. See my concerns below.

   The major argument/results presented in section 3.2 to support the conclusion "the ambient aerosol RRI is highly related with the aerosol effective density ($\rho_{eff}$) rather than the main chemical components" are Fig. 4, Fig. 5 and Fig. S6 (in the revised manuscript). However, RRI and chemical compositions in these comparisons were not taken from the same aerosol group. The RRI were taken from aerosol of a certain size (i.e., 300 nm) while the chemical compositions were taken from $PM_{2.5}$ either for direct comparison (Fig. 4 and Fig. S6) or for calculating the RRI (Fig. 5) (The water-soluble ions were from $PM_{2.5}$; whether ECOC measurements were from $PM_{2.5}$ or $PM_{10}$ are not clear in the text of section 2.1). In this case, even the same parameter may differ from each other. For example, one can clearly see multiple modes in the comparison of main aerosol components and RRI in Fig. S6.

Such comparison can be misleading because there is a danger that the readers might get an impression that the commonly used mixing rules in calculating refractive index wouldn't work for ambient aerosols, e.g., volume linear mixing rule, Maxwell-Garnet and Brüggemann mixing rule, partial molar refraction mixing rule, Lorentz-Lorenz mixing rule etc. (those requires information of chemical compositions in a mixed system). A direct consequence has already been shown in Fig. 5, where the authors delivered a message that Stelson's approach (Stelson, 1990) of calculating refractive index with partial molar refraction mixing rule did not work for the Taizhou case. This is unfair because the mismatch of different aerosol population in this comparison may to some (large) extend lead to the very scattered data points of the Stelson's approach in Fig. 5.

3. The authors' response about "how to use the parameterization in numerical models, i.e., what is the input and required parameters, may be required" is not adequate. Modelers understand "the effective density is the only parameter as input", but the real question is how to determine the effective density in the model, which hasn't been answered. I guess that one would still need to calculate $\rho_{eff}$ from densities of individual simulated chemical composition. This procedure however, may be hampered by the lack of density information of organic carbons and mixing state of black carbon, etc.

4. Concerning Reviewer 2's comment "Zhao et al. (2018b) seems to be still under discussion. The readers cannot trust the method only from the explanation in this manuscript.", the authors may want to refer to the work of Zhang et al. (JGR, 2018), where the method of combining DMA and SP2 to retrieve the real part of the refractive index of pure scattering aerosol particles has been proposed and published.

5. L50: "main aerosol" is duplicated.

6. L198-199: could it be that statistics of RRI at 200 and 300 nm is better that at 450 nm? Because the scattering signal of SP2 may become saturated for a large fraction of particles at 450 nm, and reduce the sample size. How were the double charged particles treated?

7. L147-151: "SSA is defined as the ratio of $\sigma_{sca}$ to $\sigma_{ext}$, which reflects concentration of the absorbing aerosol (Tao et al., 2014) to some extent. The $g$ expresses the distribution of the scattering light intensity in different directions (Zhao et al., 2018a). The $\sigma_{ext}$, SSA and $g$ are the most important three factors that influence the aerosol radiative properties in radiative calculation (Kuang et al., 2015)." L169-170: "… the difference between fn↓ and fn↑ (fn↓ - fn↑) is the downward radiative irradiance flux for aerosol-free conditions (Kuang et al., 2016)."

It does not seem to be appropriate to cite these references here, because such statements are rather classical textbook knowledge.

8. L192: change "The RRI and $\rho_{eff}$ vary…" to "The RRI varies".

9. L201-202: change "at about 15:00 in the morning" to "at about 15:00 in the afternoon" and change "at around 9:00 in the afternoon" to "at around 9:00 in the morning".

10. L214-215: "…Thus, the effective tend to increase with the increment of aerosol diameter." sounds like a broken sentence.

11. L254: "one day" instead of "one days".

12. Fig. 5: The caption is confusing and needs to be revised. "Comparison between the measured RRI and calculated RRI using the main aerosol chemical component from Stelson (1990) (in red star)…" Do the authors mean using the same aerosol chemical species those are needed for applying the Stelson's method, but the concentrations of the chemical components are still the measured ones from this study? If yes, please revise the caption. This is related to reviewer's comment about "Why do the authors compare a result with other at different time series and measurement site?"

**Reference**

Y. Liu, P. H. Daum, Relationship of refractive index to mass density and self-consistency of mixing rules for multicomponent mixtures like ambient aerosols. Journal of Aerosol Science 39, 974-986 (2008).

G. HÄNEL, The real part of the mean complex refractive index and the mean density of samples of atmospheric aerosol particles. Tellus 20, 371-379 (1968).

J. L. Hand, S. M. Kreidenweis, A New Method for Retrieving Particle Refractive Index and Effective Density from Aerosol Size Distribution Data. Aerosol Science and Technology 36, 1012-1026 (2002).

N. Tang, H. R. Munkelwitz, Water activities, densities and refractive indices of aqueous sulfates and sodium nitrate droplets of atmospheric importance. J. Geophys. Res. 99, 18801-18808 (1994).

P. Guyon et al., Refractive index of aerosol particles over the Amazon tropical forest during LBA-EUSTACH 1999. J. Aerosol Sci. 34, 883-907 (2003).

Y. Zhang et al., Sizing of Ambient Particles from a Single-Particle Soot Photometer Measurement to Retrieve Mixing State of Black Carbon at a Regional Site of the North China Plain. Journal of Geophysical Research: Atmospheres 123, 12,778-712,795 (2018).

A. W. Stelson, Urban aerosol refractive index prediction by partial molar refraction approach, ES&T, 24, 1990.

---

## Author Response (AR2)

Response to reviewer#1

*Comment: This study deals with a very interesting and important issue, i.e., parameterization of aerosol refractive index, which is essential for the estimation of aerosol direct radiative forcing. I read the manuscript and the authors' response with great interest. However, after careful evaluation, I agree with the other reviewer that this study is not suitable for publication in ACP as "it needs further analysis, reorganization, discussion and clarification to (prove) improve the confidence of the results (reviewer 1)". I will expand a bit on these issues as detailed below.*

**Reply:** We thank the anonymous reviewer's comments. However, we can not agree with some of the reviewer's main opinions about our manuscript. Her/His comments are addressed point-by-point and our responses are listed below. More importantly, she/he evaluated our manuscript with wrong conclusion from wrong data. We have no idea how she/he obtained our first-hand field data. We would like to provide her/him the raw data for re-analysis.

*Comment: Major Comments:*

*The parameterization scheme of the current study $(Re = (RRI2-1)/(RRI2+2) = 0.18reff)$ is in principle a justification / an update of the scheme proposed by Liu and Daum (2008) $(Re =(RRI2-1)/(RRI2+2) = 0.23reff0.39)$ based on a new dataset measured at Taizhou in China for 7 days in June 2018. The major concern from the other reviewers is whether the new parameterization is universal and applicable in global and climate models as suggested by the authors. I am not convinced by the author's arguments because of the following reasons.*

*1. Is the new scheme universal and better than the one from Liu and Daum (2008)?*

**Reply:** The new proposed parameterization scheme has been proved to be universal and applicable with more measurements data at PKU beside that at Taizhou in the revised manuscript. The results are shown in fig. R1(figure 10 in the manuscript). We also compared the measured and calculated RRI using the corresponding data published before (Guyon et al., 2003; Hand and Kreidenweis, 2002; Hänel, 1968;

Tang, 1996; Tang and Munkelwitz, 1994). We got an $R^2$ of 0.91, slope of 1.00 and intercept of 0.01. Therefore, our parameterization scheme works for both our field measurement data and the previous studies. Moreover, our proposed parameterization scheme is more acceptable in physics. By definition, the RRI of mono-component particle is defined as

$$\frac{RRI^2-1}{RRI^2+2} = \frac{N_A\alpha}{3M}\rho_{eff}, \qquad (1)$$

where $N_A$ is the universal Avagadro's number, $\alpha$ is the mean molecular polarizability, M is the molecular weight of the material and $\rho_{eff}$ is the mass effective density of the chemical component. When compared with equation 1, our parameterization scheme is more acceptable than that of Liu and Daum (2008). Based on above results and discussion, our parameterization scheme is universal and applicable.

[Figure]

**Figure R1.** Comparison between the measured and calculated RRI at PKU (in red circle) and Taizhou (in cyan hexagon) station. The triangle in black , red, blue and green corresponding the data from Hänel (1968), Tang (1996), Hand and Kreidenweis (2002), and Guyon et al. (2003) respectively. The black line is the 1:1 line.

*Comment: If a scheme is universal, it should not only explain one dataset, but also be applicable and compatible for other datasets. To come up with their parameterization scheme, Liu and Daum (2008) studied the relationship of refractive index to mass*

*density (index density relationship) for over 4000 pure materials and for aerosol particles. Note that, in Liu and Daum (2008), the summarized pure materials include organics, and investigated aerosol data cover aerosol samples from Amazon (Guyon et al., 2003), which is expected to constitute significant fraction of organics. Thus, it is not appropriate for the authors to make a statement "the influence of organic aerosols components on aerosol RRI is not considered in their work (L270-271)".*

**Reply:** Thanks for the comments. We agree with the reviewer that some of the description in our manuscript is not appropriately stated and we made revisions in L270-271. In Liu and Daum et al., they summarized 4000 pure materials and some aerosol data. However, the ambient aerosol particles are far from pure materials. We compared the measured and calculated RRI using their parameterizations scheme with the data before (Guyon et al., 2003; Hand and Kreidenweis, 2002; Hänel, 1968; Tang, 1996; Tang and Munkelwitz, 1994) and found our parameterization scheme behaved better than Liu and Daum's parameterization scheme as shown in fig. R2. Furthermore, the calculated RRI using the parameterization scheme of Liu et al is biased using our field measurement data as shown in Figure 8 in our manuscript. Our parameterization scheme works well for both the data from our field measurement results and from the previous studies (Guyon et al., 2003; Hand and Kreidenweis, 2002; Tang, 1996; Tang and Munkelwitz, 1994).

[Figure]

**Figure R2.** Comparison between the measured and calculated RRI with our parameterization scheme (star) and that of Liu's (hexagon) respectively, using the data from studies before (Guyon et al., 2003; Hand and Kreidenweis, 2002; Hänel, 1968; Tang, 1996; Tang and Munkelwitz, 1994).

[Figure]

**Figure 8.** Comparison between the measured RRI and calculated RRI using the main aerosol chemical component with the method of Stelson (1990) (in red star) and parameterization scheme proposed by Liu and Daum (2008) (in cyan hexagon).

*Comment: In Fig. C1, I compared the results of the current study with Liu and Daum (2008) by overlaying the data from the Taizhou site (small light blue dots) and from the PKU site (small pink dots) onto the original Fig. 3 of Liu and Daum (2008). Interestingly, the new datasets are not much different from the ones already summarized by Liu and Daum (2008), as they fit well into the data clouds within the same mass density range.*

[Figure]

*Fig. C1. Dependence of the effective refractive index (Re = (RRI2-1)/(RRI2+2)) on the effective mass density (reff) for pure materials and for aerosol particles. The new data sets from the current study (small light blue dots for Taizhou and small pink dots for PKU) overlay the original Fig. 3 of Liu and Daum (2008). The data summarized by Liu Daum (2008) includes over 4000 pure materials (small grey dots, including organics, inorganics and minerals, www.knovel.com), as well as ambient aerosol and lab generated surrogate with chemical compositions representing ambient aerosols (big black triangles) (Hänel, 1968; Tang and Munkelwitz, 1994; Hand and Kreidenweis, 2002; Guyon et al. 2003).*

**Reply:** The results which the anonymous reviewer provided in Fig.C1 are not acceptable. She/He plotted and analyzed with the wrong data. We would like to provide her/him the raw dataset to re-evaluate our manuscript. We are wondering how she/he obtained our firsthand field data which is not available to other groups.

The data from Liu and Daum comes from 4000 pure materials. However, ambient aerosol particles are far from pure materials. The chemical compositions of ambient aerosols are complicated including inorganics, organics and black carbon. Most of the mentioned pure materials in Liu and Daum (2008) rarely exist in the ambient aerosol. Thus, most of the pure materials have negligible influence on the ambient aerosol RRI. In dealing with the ambient RRI calculation, it is not proper to consider all the 4000 pure materials, which may leading to great uncertainties. Our measured data representing the ambient aerosol properites fit into the data clouds in Liu and Daum because there were so many inappropriate data in their figure.

*Comment: For the whole data population, it appears that Liu and Daum's scheme (black solid line) is still the best approach to describe the overall index-density relationship.*

**Reply:** We do not agree with the reviewer's comments and the reasons are listed as following:

1. In the figure 8 of our manuscript, the parameterization scheme proposed by Liu and Daum failed to describe the relationship between the effective density and RRI of our measured data.

2. Liu and Daum used many data with effective density larger than 2.0 g/cm$^3$. Many studies found that the ambient aerosol effective density was lower than 2.0g/cm$^3$ (Hand and Kreidenweis, 2002; Lee et al., 2009; Ma et al., 2017; Qiao et al., 2018; Rissler et al., 2014; Wang et al., 2010). Only the measured results in Hänel (1968) reported that the ambient aerosol density may larger than 2.0g/cm$^3$. However, they mentioned that "It has been estimated that the systematic errors can reach the magnitude of the measuring errors" using their proposed method. These data with effective density larger than 2.0 g/cm$^3$ should not be used to represent the ambient aerosol properties.

3. Liu and Daum also used the data of 4000 pure materials. However, ambient aerosol particles are far from pure materials. The chemical compositions of ambient aerosols are complicated including inorganics, organics and black carbon. Most of the mentioned pure materials in Liu and Daum (2008) rarely exist in the ambient aerosol. Thus, most of the pure materials have negligible influence on the ambient aerosol RRI. In dealing with the ambient RRI calculation, it is not proper to consider all the 4000 pure materials, which may leading to great uncertainties.

Therefore, the parameterization scheme from Liu et at may not applicable for the ambient aerosol because their results are significantly influenced by the inappropriate data.

*Comment: Because the new parametrization from the current study (blue solid line) is not able to represent the general trend in the existing dataset (over 4000 pure materials marked by small grey dots), especially it failed to explain the aerosol data from early field campaign/laboratory studies (marked by big black triangles).*

**Reply:** As mentioned above, the data from Liu and Daum comes from 4000 pure materials. However, ambient aerosol particles are far from pure materials. It is not proper to consider all the 4000 pure materials, which may leading to great uncertainties.

The reviewer concluded that our parameterization failed to explain the aerosol data from early field campaign. However, our proposed parameterization scheme works well for both our field measurement data and the early field campaign/laboratory studies as shown in fig. R1 and fig. R2. We added the comparison in the revised manuscript.

*Comment: Depending on to which degree one would like the schemes to represent the variability, for the Taizhou site the predicted average RRI (~1.44) by Liu and Daum's scheme is in a reasonable agreement with the observed 28-day average RRI of 1.425, 1.435 and 1.47 for 200 nm, 300 nm and 450 nm particles, respectively, which is probably already good enough for global and climate model applications.*

**Reply:** From fig. 8 in our manuscript, the parameterization scheme of Liu failed to fit our measured data as it biased from 1:1 line. Our parameterization scheme is better than that of Liu et al. because it works well for both our field measurement data and the early field campaign/laboratory studies as shown in fig. R1 and fig. R2.

*Comment: On the other hand, when speaking of explaining the detailed temporal and spatial variability, the prediction of the new parameterization at the PKU site is quite scattered with y = 1.0x and R2 = 0.03 (see my Major Comments 2). For example, a prediction of RRI ~1.5 with the new parameterization scheme at the PKU site may*

*correspond to a variability of real/observed RRI from 1.42 to 1.58 (Fig. C2-A).*

[Figure]

*Fig. C2. Comparison of measured and predicted RRI by the parameterization scheme of Zhao et al. (ACPD). A, my re-calculation with data from Fig. 6 of Zhao et al. (ACPD). B, the original Fig. 6 of Zhao et al. (ACPD).*

**Reply:** Again, the reviewer plotted Fig.C2 with wrong data and the results are questionable. We would like to provide the raw data for her/him to re-analysis. Also, in the revised manuscript, we added more experiment datasets at PKU site to validate our proposed parameterization scheme.

*Comment: Thus, it is unlikely that the new parameterization scheme from the current study is universal and applicable to global and climate models. In my opinion, the intrinsic scattering of the index-density relationship (Fig. C1) implies that a perfect parameterization may not be even possible. If a compromise has to be made, Liu and Daum's scheme still seems to be optimal choice in terms of universality.*

**Reply:** Based on our responses above, we do not agree with the reviewer's comment. Our parameterization scheme works well for both our own measured data and the previous published data. The parameterization scheme of Liu fails to repeat our field measurement data.

*Comment: Consistence between the PKU and Taizhou sites?*

*When comparing the consistency of different dataset, I find that I cannot reproduce*

*the results of Fig. 6 in the revised manuscript (also shown here as Fig. C2-B).*

*While the authors provided a R2 of 0.47 for the PKU site (Fig. C2-B), using the same fit function (y = ax by forcing intercept = 0), I received a coefficient of determination (R2) of only 0.03 for the same dataset (Fig. C2-A), not sufficient to support the authors' argument about consistence between the PKU and Taizhou sites. Apparently, the authors have selected the "good" slope (1.0) of y = ax and the "better" R2 (0.47) of y = ax + b to justify the advantage of their method in Fig. 6 and relevant text. This is misleading. Such a way of selectively presenting results is a serious issue and has to be corrected.*

**Reply:** We double checked our data and results. It is not acceptable that the reviewer evaluated our work with wrong dataset. Her/His results are questionable. We have no idea how the reviewer get our firsthand dataset. Anyway, we would like to provide her/him the raw dataset for re-analysis.

*Comment: Other Comments:1. Abstract: The retrieved RRI is for pure scattering aerosols (or may be extended for the coating materials when calculating the effective refractive index of mixed black particles?), while the effective density is measured for all aerosols (both scattering and absorbing aerosols). Direct comparison between the two measured quantifies may induce uncertainties, and should be justified or at least clarified.*

**Reply:** Thanks for the comments. The following discussions would prove that the measured effective density corresponding to these of scattering aerosols. The effects of absorbing aerosols can be neglected.

Fig. R3 gave three examples of the aerosol PNSDs that had passed the CPMA and were measured by the SMPS. The mass values of the aerosol that can pass through the CPMA were set to be 12 fg, 1 fg and 1.4 fg respectively. From fig. R3, the aerosols that passed through the CPMA were mainly composed of three modes. For each mode, the aerosol number concentrations were fit by the log-normal distribution function:

$$\mathrm{N(H)} = \frac{N_0}{\sqrt{2\pi}\log(\sigma_g)} \cdot exp\left[-\frac{\log Dp - \log(Dp)}{2\log^2(\sigma_g)}\right] \qquad (2)$$

Where $\sigma_g$ is the geometric standard deviation; $Dp$ is the geometric mean diameter and $N_0$ is the number concentrations for a peak mode. The geometric mean diameter is further analyzed.

We would demonstrate that the mode 1, 2 and 3 in the figure correspond to those aerosols of absorbing aerosol, scattering aerosol, and scattering aerosol with double charges respectively.

Based on the principle of CPMA, when the CPMA is selecting the aerosols at mass $m_0$ of single charged aerosol particles. Then theses multiple-charged (numbers of charges is n) aerosol particles with mass concentration of $nm_0$ can pass through the CPMA at the same time. We assume that the geometric diameter of the single charge aerosol particles is $D_0$, and the effective density among different aerosol diameter doesn't have significant variations. Then the geometric diameter of the multiple charged aerosol particles is $\sqrt[3]{n}$m.

As for the DMA, when a voltage (*V*) is applied to the DMA, only a narrow size range of aerosol particles, with the same electrical mobility ($Z_p$) can pass through the DMA (Knutson and Whitby, 1975). The $Z_p$ is expressed as:

$$Z_P = \frac{Q_{sh}}{2\pi VL} ln(\frac{r_1}{r_2}) \qquad (3)$$

where $Q_{sh}$ is the sheath flow rate; *L* is the length of the DMA; $r_1$ is the outer radius of annular space and $r_2$ is the inner radius of the annular space. The transfer function refers to the probability that a particle with a certain electrical mobility can pass through the DMA. For a given *V*, the transfer function is triangular-shaped, with the peaking value of 100% and a half width (HW) of

$$\Delta Z_p = Z_P \frac{Q_a}{Q_{sh}} \qquad (4)$$

The aerosol $Z_p$, which is highly related to the aerosols diameter ($D_p$) and the number of elementary charges on the particle (*n*), is defined as:

$$Z_p = \frac{neC(D_p)}{3\pi\mu D_p} \qquad (5)$$

where *e* is the elementary charge; $\mu$ is the gas viscosity coefficient, $C(D_p)$ is the

Cunningham slip correction that is defined by:

$$C = 1 + \frac{2\tau}{D_p}(1.142 + 0.558e^{-\frac{0.999D_p}{2\tau}}) \qquad (6)$$

where $\square\square$ is the gas mean free path.

Therefore, the corresponding electrical diameter Zp(n) of the particles with n charges and diameters $\sqrt[3]{n}$m can be calculated based on equation 5. The diameter (Dn) measured by the DMA can be calculated with electrical diameter Zp(n) and single charged particle by using equation 5 again. The relations ship of the Dn and the aerosol diameter selected by the DMA can be determined by changes the aerosol Dp and charge numbers. The results are shown in fig. R4.

[Figure]

**Figure R3.** The measured aerosol PNSD (black dotted line), fit aerosol number PNSD (blue solid line), and fit aerosol PNSD at three different mode in different colors. Panel (a) (b) (c) corresponding to the aerosol mass concentrations of 12, 1, 1.45 fg.

[Figure]

**Figure R4.** The relationship between the measured diameter by the DMA and the calculated true aerosol diameter of different charges in the CPMA-SMPS system.

The fit geometric diameter of mode 2 and mode 3 were also plot in fig. R4. From fig. R4, the measured relationships of the mode 2 and mode 3 agree well with that of double charged diameters. The deviations might resulted from the assumptions that the aerosol effective density doesn't change among different diameters. Therefore, we conclude that the mode 3 corresponds to the double-charged aerosols and should not be used to analyze the effective density.

However, the mode 1 and mode 2 corresponding to the effective densities around 1.0 g/cm$^3$ and 1.5 g/cm$^3$ respectively. Previous studies had shown that the ambient BC aerosol is chain like in the morphology and have smaller effective density values. At the same time, the fit aerosol number concentrations of mode one is only between 1/5 to 1/3 of the mode two. Based on the size-selected aerosol properties, there were only mean 25% percent of the ambient aerosols that contains BC (another paper in preparation). Therefore, the mode 1 corresponds to the BC-contained aerosols and mode 2 corresponds mainly to scattering aerosols.

In our study, the effective density correspond to the geometric diameters of mode 2 was used. Thus, both the measured aerosol effective density and RRI correspond to these scattering aerosols. We added the above discussion in the revised manuscript.

*Comment: 2. Abstract and section 3.2: I suggest to remove "rather than the main chemical components" from "We find that the ambient aerosol RRI is highly related with the aerosol effective density (reff) rather than the main chemical components", or change'related' to 'correlated'. This is because both refractive index and effective density are determined by main chemical components of aerosol particles. Even for the proposed application in global or climate model (calculation of RRI from ρeff), one would still need the simulated chemical compositions to calculate ρeff (see my Other Comments 3).*

**Reply:** Thanks for the comments. We have revised the manuscript.

*Comment: Along this line, section 3.2 should be substantially revised by including more detailed and thorough evidences and discussions. See my concerns below. The major argument/results presented in section 3.2 to support the conclusion "the ambient aerosol RRI is highly related with the aerosol effective density (reff) rather than the main chemical components" are Fig. 4, Fig. 5 and Fig. S6 (in the revised manuscript). However, RRI and chemical compositions in these comparisons were not taken from the same aerosol group. The RRI were taken from aerosol of a certain size (i.e., 300 nm) while the chemical compositions were taken from PM2.5 either for direct comparison (Fig. 4 and Fig. S6) or for calculating the RRI (Fig. 5) (The water-soluble ions were from PM2.5; whether ECOC measurements were from PM2.5 or PM10 are not clear in the text of section 2.1). In this case, even the same parameter may differ from each other. For example, one can clearly see multiple modes in the comparison of main aerosol components and RRI in Fig. S6.*
*Such comparison can be misleading because there is a danger that the readers might get an impression that the commonly used mixing rules in calculating refractive index wouldn't work for ambient aerosols, e.g., volume linear mixing rule, Maxwell-Garnet and Brüggemann mixing rule, partial molar refraction mixing rule, Lorentz-Lorenz mixing rule etc. (those requires information of chemical compositions in a mixed system). A direct consequence has already been shown in Fig. 5, where the authors delivered a message that Stelson's approach (Stelson, 1990) of calculating refractive*

*index with partial molar refraction mixing rule did not work for the Taizhou case. This is unfair because the mismatch of different aerosol population in this comparison may to some (large) extend lead to the very scattered data points of the Stelson's approach in Fig. 5.*

**Reply:** Thanks for the comments. The ECOC measurements were from PM2.5. We have changed the manuscript.

We would first demonstrate that the measured RRI at a given diameter of 250 nm is in consistent with that of the bulk aerosol optical properties derived RRI. The aerosol-effective RRI was retrieved by applying the Mie scattering theory to the aerosol particle number size distribution (PNSD), aerosol bulk scattering coefficient and aerosol absorbing coefficient data (Cai et al., 2011). Results in fig. R3 show that the measured and calculated RRI shows good consistence.

[Figure]

**Figure. R3.** Comparison between the measured RRI at 250 nm and the calculated RRI using the aerosol bulk aerosol optical properties.

Therefore, the size-resolved aerosol RRI can be used to some extent represent the bulk aerosol optical properties. The measured RRI at 250 nm and calculated aerosol RRI using the bulk aerosol main chemical composition should to some extent correlated with each other. However, as shown in fig. 5 in the manuscript, the measured RRI at 250 nm and calculated RRI using the method Stelson (1990) has $R^2$ of 0.07. Therefore, the ambient aerosol RRI calculated from bulk aerosol main inorganic component may lead to great uncertainties.

The commonly used mixing rules in calculating refractive index may not work for ambient aerosols. The main reason of the discrepancy are due to ignoring influence of organic. The manuscript was revised correspondingly.

*Comments: 3. The authors' response about "how to use the parameterization in numerical models, i.e., what is the input and required parameters, may be required" is not adequate. Modelers understand "the effective density is the only parameter as input", but the real question is how to determine the effective density in the model, which hasn't been answered. I guess that one would still need to calculate reff from densities of individual simulated chemical composition. This procedure however, may be hampered by the lack of density information of organic carbons and mixing state of black carbon, etc.*

**Reply:** Thanks for the comment. The motivation of our research is to bridge the gap between the ambient aerosol RRI and ambient aerosol effective density. How to determine the effective density in the model is out of the scope of our research. In fact, there are many methods available to estimate the ambient aerosol effective density (Hu et al., 2012; Karg, 2000; Qiao et al., 2018; Schmid et al., 2007; Zhang et al., 2016). The RRI can be calculated conveniently with available effective density using our parameterization scheme.

*Comments:4. Concerning Reviewer 2's comment "Zhao et al. (2018b) seems to be still under discussion. The readers cannot trust the method only from the explanation in this manuscript.", the authors may want to refer to the work of Zhang et al. (JGR, 2018), where the method of combining DMA and SP2 to retrieve the real part of the refractive index of pure scattering aerosol particles has been proposed and published.*

**Reply:** Thanks for the comment. The work of Zhao et al. (2018b) has been published as Zhao et al. (2019). The details of combining DMA and SP2 to retrieve the real part of the refractive index can be found in that.

*Comment: 5. L50: "main aerosol" is duplicated.*

**Reply:** Thanks for the comments. We have revised the manuscript.

*Comment: 6. L198-199: could it be that statistics of RRI at 200 and 300 nm is better that at 450 nm? Because the scattering signal of SP2 may become saturated for a large fraction of particles at 450 nm, and reduce the sample size. How were the double charged particles treated?*

**Reply:** Thanks for the comments. We checked the scattering signals and the scattering signal of SP2 is not saturated for these particles at 450 nm. The reason of RRI get more dispersed at 450 nm might be related the complicated aging processing and sources of these large particles compared with those at 200 nm and 300 nm. The method of treating the double charged particles are detailed in Zhao et al. (2019).

*Comment: 7. L147-151: "SSA is defined as the ratio of ssca to sext, which reflects concentration of the absorbing aerosol (Tao et al., 2014) to some extent. The g expresses the distribution of the scattering light intensity in different directions (Zhao et al., 2018a). The sext, SSA and g are the most important three factors that influence the aerosol radiative properties in radiative calculation (Kuang et al., 2015)."*
*L169-170: "… the difference between fn⁻ and fn( fn⁻ - fn) is the downward radiative irradiance flux for aerosol-free conditions (Kuang et al., 2016)."*
*It does not seem to be appropriate to cite these references here, because such statements are rather classical textbook knowledge.*

**Reply:** Thanks for the comment. We have revised the manuscript.

*Comment: 8. L192: change "The RRI and reff vary…" to "The RRI varies".*

**Reply:** Thanks for the comment. We have revised the manuscript.

*Comment: 9. L201-202: change "at about 15:00 in the morning" to "at about 15:00 in the afternoon" and change "at around 9:00 in the afternoon" to "at around 9:00 in the morning".*

**Reply:** Thanks for the comment. We have revised the manuscript.

*Comment: 10. L214-215: "…Thus, the effective tend to increase with the increment of aerosol diameter." sounds like a broken sentence.*

**Reply:** Thanks for the comment. We have deleted the corresponding content in the manuscript.

*Comment: 11. L254: "one day" instead of "one days".*

**Reply:** Thanks for the comment. We have revised the manuscript.

*Comment: 12. Fig. 5: The caption is confusing and needs to be revised. "Comparison between the measured RRI and calculated RRI using the main aerosol chemical component from Stelson (1990) (in red star)…" Do the authors mean using the same aerosol chemical species those are needed for applying the Stelson's method, but the concentrations of the chemical components are still the measured ones from this study? If yes, please revise the caption. This is related to reviewer's comment about "Why do the authors compare a result with other at different time series and measurement site?"*

**Reply:** Thanks for the comments. We have revised the manuscript.

Cai, Y., Montague, D.C., Deshler, T. (2011) Comparison of measured and calculated scattering from surface aerosols with an average, a size-dependent, and a time-dependent refractive index. Journal of Geophysical Research 116.

Guyon, P., Boucher, O., Graham, B., Beck, J., Mayol-Bracero, O.L., Roberts, G.C., Maenhaut, W., Artaxo, P., Andreae, M.O. (2003) Refractive index of aerosol particles over the Amazon tropical forest during LBA-EUSTACH 1999. Journal of Aerosol Science 34, 883-907.

Hand, J.L., Kreidenweis, S.M. (2002) A New Method for Retrieving Particle Refractive Index and Effective Density from Aerosol Size Distribution Data. Aerosol Science And Technology 36, 1012-1026.

Hänel, G. (1968) REAL PART OF MEAN COMPLEX REFRACTIVE INDEX AND

MEAN DENSITY OF SAMPLES OF ATMOSPHERIC AEROSOL PARTICLES. Tellus 20, 371-&.

Hu, M., Peng, J., Sun, K., Yue, D., Guo, S., Wiedensohler, A., Wu, Z. (2012) Estimation of size-resolved ambient particle density based on the measurement of aerosol number, mass, and chemical size distributions in the winter in Beijing. Environ Sci Technol 46, 9941-9947.

Karg, E. (2000) The density of ambient particles from combined DMA and APS data. Journal of Aerosol Science 31, 759-760.

Knutson, E.O., Whitby, K.T. (1975) Aerosol classification by electric mobility: apparatus, theory, and applications. Journal of Aerosol Science 6, 443-451.

Lee, S.Y., Widiyastuti, W., Tajima, N., Iskandar, F., Okuyama, K. (2009) Measurement of the Effective Density of Both Spherical Aggregated and Ordered Porous Aerosol Particles Using Mobility- and Mass-Analyzers. Aerosol Science And Technology 43, 136-144.

Liu, Y., Daum, P.H. (2008) Relationship of refractive index to mass density and self-consistency of mixing rules for multicomponent mixtures like ambient aerosols. Journal of Aerosol Science 39, 974-986.

Ma, Y., Li, S., Zheng, J., Khalizov, A., Wang, X., Wang, Z., Zhou, Y. (2017) Size-resolved measurements of mixing state and cloud-nucleating ability of aerosols in Nanjing, China. Journal of Geophysical Research: Atmospheres 122, 9430-9450.

Qiao, K., Wu, Z., Pei, X., Liu, Q., Shang, D., Zheng, J., Du, Z., Zhu, W., Wu, Y., Lou, S., Guo, S., Chan, C.K., Pathak, R.K., Hallquist, M., Hu, M. (2018) Size-resolved effective density of submicron particles during summertime in the rural atmosphere of Beijing, China. Journal of Environmental Sciences.

Rissler, J., Nordin, E.Z., Eriksson, A.C., Nilsson, P.T., Frosch, M., Sporre, M.K., Wierzbicka, A., Svenningsson, B., Londahl, J., Messing, M.E., Sjogren, S., Hemmingsen, J.G., Loft, S., Pagels, J.H., Swietlicki, E. (2014) Effective density and mixing state of aerosol particles in a near-traffic urban environment. Environ Sci Technol 48, 6300-6308.

Schmid, O., Karg, E., Hagen, D.E., Whitefield, P.D., Ferron, G.A. (2007) On the effective density of non-spherical particles as derived from combined measurements of aerodynamic and mobility equivalent size. Journal of Aerosol Science 38, 431-443.

Stelson, A.W. (1990) Urban aerosol refractive index prediction by partial molar refraction approach. Environ.sci.technol 24:11, 1676-1679.

Tang, I.N. (1996) Chemical and size effects of hygroscopic aerosols on light scattering coefficients. Journal of Geophysical Research: Atmospheres 101, 19245-19250.

Tang, I.N., Munkelwitz, H.R. (1994) WATER ACTIVITIES, DENSITIES, AND REFRACTIVE-INDEXES OF AQUEOUS SULFATES AND SODIUM-NITRATE DROPLETS OF ATMOSPHERIC IMPORTANCE. Journal Of Geophysical Research-Atmospheres 99, 18801-18808.

Wang, X., Zhang, L., Moran, M.D. (2010) Uncertainty assessment of current size-resolved parameterizations for below-cloud particle scavenging by rain. Atmospheric Chemistry and Physics 10, 5685-5705.

[revised manuscript text omitted]

**1 The daily variation and probability distribution of the ρ_eff**

[Figure]

**Figure. S1.** Daily variations of the ρ_eff (a), (c) (e), and the probability distribution of the measured ρ_eff (b), (d) (f) for the (a), (b) 200 nm, (c), (d) 300 nm, and (e), (f) 400nm aerosol.

**2. Comparison the measured RRI Aerosol Components**

[Figure]

**Figure S2.** Comparison the measured RRI at 300nm with the measured (a) $\sigma_{sca}$ at 525nm, mass concentrations of (b) OM, (c) $SO_4^{2-}$, (d) $Cl^-$, (e) $NH_4^+$ and (f) $NO_3^-$.

---

## Author Response (AR3)

Response to reviewer#1

Thanks for the reviewer's valuable comments! The point-by-point responses are listed below.

**Comment:** This study provides a novel approach of deriving the real part of refractive index (RRI) for non-absorbing particles using single particle technique, and uses the measured effective density to parameterize this parameter for the ambient. The article is well structured and the data has been analyzed carefully, I would recommend publication after addressing the following points:

**Reply:** We thank the anonymous reviewer's positive comments.

**Comment:** 1)Regarding the representativeness of the RRI vs ED relationship, I wouldn't think this has to be necessarily consistent at all sites or under all environments. This is because of the variabilities of particle morphology and mixing states among compositions. A number of urban studies have been given by the authors, it is sufficient to demonstrate the broad implications of this parameterization at least for urban environment, which needs to be mentioned in the texts though.

**Reply:** We agree with the reviewer and have revised the title and abstract according to the reviewer's comment. The aerosols mentioned in our manuscript related to the urban aerosols.

**Comment:* 2)*Have you performed CPMA inversion for the CPMA-derived size distribution, such as in Fig. 3?**

**Reply:** Thanks for the comment. In our work, the inversion was made for the DMA-CPC derived size distribution. Performing the CPMA inversion for the CPMA-derived size distribution is not necessary in this work. It is the geometric mean diameter of each mode as shown in fig. 3 that matters to our results of deriving the effective density.

*Comment:* 3)I would like to see more details about how the RI has been calculated from the SP2 measurement, i.e. the scattering intensity distribution at a range of SMPS-selected sizes, what is the minimum threshold of acceptable scattering signal, and the uncertainty of scattering intensity at different sizes.

**Reply:** Thanks for the comment. The details of deriving the RRI from DMA-SP2 system has been discussed in Zhao et al. (2019). In Zhao et al. (2019), the minimum threshold of acceptable scattering signal would change with different instrument. The lower detecting limit of scattering signal should be the scattering signals of ammonium sulfate with diameter around 170 nm. As discussed in section 4.2.1 in Zhao et al. (2019), the uncertainties of scattering intensity at different sizes should be around 6.8% for different size. The uncertainties may be different for different instrument. For more information, please refer to Zhao et al. (2019) (Zhao, G., Zhao, W., Zhao, C. (2019) Method to measure the size-resolved real part of aerosol refractive index using differential mobility analyzer in tandem with single-particle soot photometer. Atmospheric Measurement Techniques 12, 3541-3550).

*Comment:* 4)*Did you use the solid angle integration of SP2 detected scattering by considering the planes of both in parallel with aerosol jet and laser beam, as detailed in (Moteki and Kondo, 2007).*

**Reply:** In our work, the scattering signals is calculated by using

$$S_1 = C_0 \cdot I_0 \cdot \sigma \cdot (PF_{45^o} + PF_{135^o}), \qquad (1)$$

where  $I_0$  is the laser's intensity;  $\sigma$  is the scattering coefficient of the sampled aerosol,  $PF_{45^o}$  and  $PF_{135^o}$  are scattering phase function at 45° and 135° respectively of the sampled aerosols; and  $C_0$  is a constant that is determined by the distance from the aerosol to the APD and the area of the APD.

The scattering signal can be calculated by

$$S_2 = \frac{C_0 \cdot I_0 \cdot \int_{\Omega} \sigma_{\Omega} \rho F_{\Omega} d\Omega}{\int_{\Omega} \sigma_{\Omega} d\Omega},$$
 (2)

Where the  $\Omega$  is pre-determined solid angle where the PMT can receive the aerosol scattering signals. S1 is a simplified condition of S2.

We calculated the  $S_2$  values of different solid angle range when the angle of the  $\theta$  is in the range of 35~55° and 125~145°. It is found that the  $S_2/S_1$  is always in the range of 0.97~1.03 for aerosol diameter range of 200 nm and 500 nm. The uncertainties related with using equation 1 instead of 2 is less than 3%. Therefore, the solid angle of the PMT had little influence on our derived RRI as uncertainties of scattering intensity measured by PMT is 6.8% (Zhao et al., 2019).

**Comment:** 5) According to the analysis, particle morphology is way more dominant than other factors. This is an important message. Would you like to comment how the mobility dimeter as sized by DMA could affect the particle sizing technique, i.e. to convert the mobility size to volume-equivalent size, then affecting the derived RRI on a mobility size basis? Will this create a necessary link/relationship between the RRI and ED?

**Reply:** Thanks for the comments. Many closure studies between the measured and calculated aerosol optical properties using Mie scattering theory validated the sphericity of the ambient continental aerosols (Chen et al., 2014; Ma et al., 2014; Ma et al., 2011; Wex et al., 2002). Based on these studies, it is applicable that these particles are spherical for accumulation mode aerosols.

**Comment:* 6)Once the RRI is parameterized, how to work out the optical properties by assuming particle is sphere again?**

**Reply:** Thanks for the comment. The aerosol optical properties can be calculated using the Mie scattering model using the given parameterized RRI and size-distribution of ambient aerosols (Bohren and Huffman, 2007).

Bohren, C.F., Huffman, D.R., (2007) Absorption and Scattering by a Sphere, Absorption and Scattering of Light by Small Particles. Wiley-VCH Verlag GmbH, pp. 82-129.

Chen, J., Zhao, C.S., Ma, N., Yan, P. (2014) Aerosol hygroscopicity parameter derived from the light scattering enhancement factor measurements in the North China Plain.

Atmos. Chem. Phys. 14, 8105-8118.

Ma, N., Birmili, W., Müller, T., Tuch, T., Cheng, Y.F., Xu, W.Y., Zhao, C.S., Wiedensohler, A. (2014) Tropospheric aerosol scattering and absorption over central Europe: a closure study for the dry particle state. Atmospheric Chemistry and Physics 14, 6241-6259.

Ma, N., Zhao, C.S., Nowak, A., Müller, T., Pfeifer, S., Cheng, Y.F., Deng, Z.Z., Liu, P.F., Xu, W.Y., Ran, L., Yan, P., Göbel, T., Hallbauer, E., Mildenberger, K., Henning, S., Yu, J., Chen, L.L., Zhou, X.J., Stratmann, F., Wiedensohler, A. (2011) Aerosol optical properties in the North China Plain during HaChi campaign: an in-situ optical closure study. Atmos. Chem. Phys. 11, 5959-5973.

Wex, H., Neusüß, C., Wendisch, M., Stratmann, F., Koziar, C., Keil, A., Wiedensohler, A., Ebert, M. (2002) Particle scattering, backscattering, and absorption coefficients: An in situ closure and sensitivity study. Journal of Geophysical Research: Atmospheres 107, LAC 4-1-LAC 4-18.

Zhao, G., Zhao, W., Zhao, C. (2019) Method to measure the size-resolved real part of aerosol refractive index using differential mobility analyzer in tandem with single-particle soot photometer. Atmospheric Measurement Techniques 12, 3541-3550.

Response to reviewer#2

Thanks for the reviewer's helpful comments! The point-by-point responses are listed below.

**Comment:** This study designed a field measurement system and found a new method to better calculate the real part of refractive index where information of aerosol density is available. The topic of the study is undoubtedly of high scientific and practical importance. On the whole, the experimental methodology and data analyzing procedures look to be correct and the findings will significantly improve the estimation of aerosol radiative forcing. Therefore, this manuscript is helpful for the audience of atmospheric chemistry and physics, but not without a major revision. Some comments and suggestions are listed below.

**Reply:** We thank the anonymous reviewer's comments.

**Comment:** Main Points (1) I am a little concerned about the title "A new parameterization scheme of …". For a parameterization scheme, parameters should be changeable at different situation. In this study, a coefficient of 0.18 is obtained from two field measurements and should be applicable to the polluted regions. But as shown in equation 2, the definition of this coefficient suggests that it would vary with the molecular polarizability and molecular weight of the aerosols composition. As a result, I am not sure whether the coefficient 0.18 would be applicable for sea salt aerosols or organic-dominated aerosols. An alternative way is to limit the scope to a certain kind of atmospheric condition. By the way, is the parameter derived from the two measurements exactly the same?

**Reply :** Thanks for the comment. We have revised the title and abstract correspondingly. The aerosols mentioned in our manuscript related to the urban aerosols.

The derived parameter from the two measurement sites were slightly different. The parameter is 0.182 for Taizhou site and 0.187 for PKU site. The number of effective data in Taizhou is more than that of PKU. Thus, we chose the coefficient 0.18 in our research.

**Comment:** (2) The usage of "RRI" or its related form in the paper is always confusing. Whether it is measured or calculated, whether it is size-resolved or not. It is suggested to double check the usage of "RRI" or its related form all through the paper. A clear parameterization table with definition would be helpful for readership to better understand the paper. Also, why the authors use different size-resolved RRI at different places? I think there are size-resolved RRI at 200nm, 250nm and 300nm in different discussion.

**Reply:** Thanks for the comment. We have added a table to explain the abbreviations in our supplementary material to help understand the paper. Some descriptions in the manuscript were revised.

The size-resolved RRI at 200 nm, 300 nm and 450 nm were discussed in the new manuscript. We have revised some of discussions correspondingly and some data were changed in the manuscript.

**Comment:** (3) The structure of the current manuscript is not well organized. For example, in the Data and Methods part, the readers will have an expression that this paper is based on one campaign in Taizhou. However, in the discussion part (line 298), the measurement data at PKU site is also used, but without any description of the measurement. Another example is that there is too much background discussion and methodology in the conclusion part.

**Reply:** Thanks for the comment. We have revised the manuscript.

**Comment:** (4) The authors may need to be more careful on some statements made in the manuscript. For example, line 344, "Our proposed parameterizations scheme is a perfect substitute" is not appropriate for a scientific paper. Also, line 258 "the RRI tend to increase with the OM mass fraction ratio". I don't recognize clear trend in fig 7. A simple hypothesis testing may be needed here. **Reply:** Thanks for the comment. We have revised the manuscript. We agree with the reviewer that there was no clear relationship between the OM mass fraction ration and RRI after a simple hypothesis testing. Some of the corresponding discussions were removed from the manuscript.

**Comment:** (5) The mode 1, 2, 3 derived from DMA-CPMA-CPC measurement are considered as light absorbing aerosols, scattering aerosols and double charged aerosols. Though the aerosols with lower density are very likely the fresh emitted light absorbing aerosols, those with higher density could also be fully aged light absorbing aerosols. In my opinion, mode 1 is more like "fractal aerosols" and mode 2 is "compact aerosols". This definition may not influence the final conclusion, but still need to be carefully discussed. One suggestion is to compare the aerosol number in Mode 1 and BC number concentration measured by SP2 at different size to make sure they are comparable.

**Reply:** Thanks for the comment. We agree with the reviewer's idea that the aerosols in mode 2 correspond to the compact aerosols. Previous studies have shown that the ambient BC aerosol was chain like in the morphology and had smaller effective density values (Peng et al., 2016). At the same time, the fit aerosol number concentrations of mode one is only between 1/5 to 1/3 of the mode two. Based on the size-selected aerosol properties measured by the SP2, there were only mean 25% percent of the ambient aerosols that contains BC. Therefore, the mode 1 and mode 2 corresponded to the BC-contained aerosols and scattering aerosols respectively. There were some compacted BC-contained aerosols that may fit in mode 2. We focus on the fit geometric mean diameter of mode 2, which corresponding to the scattering aerosols that dominated this mode. Therefore, these compacted BC aerosols would not influence our final conclusion.

We made some revisions in the manuscript.

**Comment:** (6) Line 20, the authors stated "For the first time, the size-resolved ambient aerosol RRI and peff are measured simultaneously by our designed

measurement system". Since the particle size (also chemical compositions) is linked to distinct formation processes and stages of haze development, such as nucleation and growth from clean, transition, to polluted periods (Guo et al., Elucidating severe urban haze formation in China, Proc. Natl. Acad. Sci. USA 111, 17373, 2014; Wang et al., Persistent sulfate formation from London Fog to Chinese Haze, Proc. Natl. Acad. Sci. USA 113, 13630, 2016), it would be necessary that a connection between the RRI and haze development is identified.

**Reply:** Thanks for the comment. In this work, we focus on the parameterization scheme of the ambient aerosol RRI. The reviewer provide a helpful suggestion for our future work. More work will be carried out to identify the relationship of RRI and haze development.

*Comment:* (7) I also believe that some references in this paper were outdated, and a significant effort is needed to address such. Below are some examples.

Line 26, The author stated that "Atmospheric aerosols can significantly influence the regional air quality and climate system by scattering and absorbing the solar radiation (Seinfeld et al., 1998)". Several other most recent papers on this topic need to be discussed (i.e., An et al., Severe haze in Northern China: A synergy of anthropogenic emissions and atmospheric processes, Proc. Natl. Acad. Sci. USA 116, 8657, 2019; Zhang et al., Formation of urban fine particulate matter, Chem. Rev. 115, 3803, 2015; Wang et al., Light absorbing aerosols and their atmospheric impacts, Atmos. Environ. 81, 713, 2013).

**Reply:** Thanks for the comment. We made some revisions in the manuscript.

Comment: Technical comments Line 13, delete "Mainly"

**Reply:** Thanks for the comment. We have revised it.

Comment: Line 15, change "Results" to "The results"

**Reply:** Thanks for the comment. We have revised it.

*Comment:* Line 16, the sentence "vary by 40% corresponding to the variation of the measured aerosol RRI" is confusing.

**Reply:** Thanks for the comment. We have revised the sentence. The direct aerosol radiative forcing is estimated to vary by 40% when the RRI were varied between 1.36 and 1.56.

**Comment: Line 19, delete "schemes"**

**Reply:** Thanks for the comment. We have revised it.

*Comment: Line 173, "relations ship" should be "relationship"*

**Reply:** Thanks for the comment. We have revised it.

*Comment:* Line 301, "equation 7" should be the equation 9?

**Reply:** Thanks for the comment. We have revised it.

[revised manuscript text omitted]